# Knockout of cyclin-dependent kinases 8 and 19 leads to depletion of cyclin C and suppresses spermatogenesis and male fertility in mice

Alexandra V Bruter[1,2†], Ekaterina A Varlamova[1*†], Nina I Stavskaya[3], Zoia G Antysheva[3], Vasily N Manskikh[4], Anna V Tvorogova[1], Diana S Korshunova[1], Alvina I Khamidullina[1,3], Marina V Utkina[5], Viktor P Bogdanov[6], Iuliia P Baikova[1], Alyona I Nikiforova[7], Eugene A Albert[6], Denis O Maksimov[6], Jing Li[8], Mengqian Chen[8,9], Gary P Schools[8], Alexey V Feoktistov[10], Alexander A Shtil[2,3], Igor B Roninson[8], Vladislav A Mogila[3,8], Yulia Y Silaeva[1,3], Victor V Tatarskiy[1,3]

[1]Center for Precision Genome Editing and Genetic Technologies for Biomedicine, Institute of Gene Biology, Russian Academy of Sciences, Moscow, Russian Federation; [2]Blokhin National Medical Research Center of Oncology, Moscow, Russian Federation; [3]Institute of Gene Biology, Russian Academy of Sciences, Moscow, Russian Federation; [4]Belozersky Institute of Physico-Chemical Biology, Lomonosov Moscow State University, Moscow, Russian Federation; [5]Endocrinology Research Centre, Moscow, Russian Federation; [6]Life Sciences Research Center, Moscow Institute of Physics and Technology, Dolgoprudny, Russian Federation; [7]Institute of Mitoengineering MSU, Moscow, Russian Federation; [8]Department of Drug Discovery and Biomedical Sciences, University of South Carolina, Columbia, United States; [9]Senex Biotechnology, Inc, Columbia, United States; [10]The Engelhardt Institute of Molecular Biology, Russian Academy of Sciences, Moscow, Russian Federation

*For correspondence: katerinavarlamova196@gmail.com

†These authors contributed equally to this work

## eLife Assessment

This **valuable** study reports the critical role of two cyclin-dependent kinases, CDK8 and CDK19, in spermatogenesis. The data presented are generally supportive of the main conclusion and are considered **solid**. This work may be of interest to reproductive biologists and physicians working on male fertility.

**Abstract** CDK8 and CDK19 paralogs are regulatory kinases associated with the transcriptional Mediator complex. We have generated mice with the systemic inducible *Cdk8* knockout on the background of *Cdk19* constitutive knockout. *Cdk8/19* double knockout (iDKO) males, but not single *Cdk8* or *Cdk19* KO, had an atrophic reproductive system and were infertile. The iDKO males lacked postmeiotic spermatids and spermatocytes after meiosis I pachytene. Testosterone levels were decreased whereas the amounts of the luteinizing hormone were unchanged. Single-cell RNA sequencing showed marked differences in the expression of steroidogenic genes (such as *Cyp17a1*, *Star*, and *Fads*) in Leydig cells concomitant with alterations in Sertoli cells and spermatocytes, and were likely associated with an impaired synthesis of steroids. *Star* and *Fads* were also downregulated in cultured Leydig cells after iDKO. The treatment of primary Leydig cell culture with a CDK8/19

inhibitor did not induce the same changes in gene expression as iDKO, and a prolonged treatment of mice with a CDK8/19 inhibitor did not affect the size of testes. iDKO, in contrast to the single knockouts or treatment with a CDK8/19 kinase inhibitor, led to depletion of cyclin C (CCNC), the binding partner of CDK8/19 that has been implicated in CDK8/19-independent functions. This suggests that the observed phenotype was likely mediated through kinase-independent activities of CDK8/19, such as CCNC stabilization.

## Introduction

Multiple protein complexes regulate transcription in response to developmental cues, changes in the environment, damage, and other factors, allowing for precise control of gene expression. Among such complexes, the multiprotein Mediator complex is especially important for controlling transcription, regulated by a multitude of signaling pathways through their transcription factors (TFs). The Mediator complex bridges the regulatory enhancer regions with promoters, thereby recruiting RNA polymerase II (RNA Pol II) and the preinitiation complex. The Mediator possesses its own regulatory subpart – the cyclin-dependent kinase module (CKM), containing either the CDK8 or the CDK19 protein kinases, in a complex with their binding partner, cyclin C (CCNC), as well as proteins MED12 and MED13 (*Luyties and Taatjes, 2022*). While CKM is a nuclear complex that regulates transcription, both CCNC (*Ježek et al., 2019*) and MED12 (*Zhang et al., 2020*) were shown to have CKM-independent cytoplasmic activities. The CDK8/19 kinase activity positively regulates transcription induced by different signals (*Chen et al., 2017*; *Chen et al., 2023*), at least in part by mediating the RNA Pol II release from pausing (*Steinparzer et al., 2019*). Besides, CDK8 and CDK19 directly phosphorylate a number of transcription factors such as STATs (*Martinez-Fabregas et al., 2020*), SMADs (*Alarcón et al., 2009*), and others, thereby modulating transcription of cognate genes. The CDK8/19 kinase activity also negatively regulates transcription, at least in part, through downregulating the protein levels of all the components of the Mediator and CKM (*Chen et al., 2023*).

Despite important CDK8/19 functions discovered in primary cell culture and cell lines (*Luyties and Taatjes, 2022*), the insights gained from in vivo models are limited. Mice with constitutively knocked out *Cdk19* (CDK19 KO) were generated as a part of the COMP project and are basically asymptomatic, fertile, and have a normal lifespan. It has been shown recently that *Cdk19^-/-* Lin^- hematopoietic stem cells divide slower (*Zhang et al., 2022*), however, the blood cell count was unaffected. Heterozygous mice with a constitutive *Cdk8* inducible knockout (CDK8 iKO) were asymptomatic but, when crossed, no homozygous pups were born (*Postlmayr et al., 2020*; *Westerling et al., 2007*). Moreover, conditional CDK8 iKO is almost asymptomatic in adult mice. Minor differences were observed in colon epithelial differentiation (*Dannappel et al., 2022*; *Prieto et al., 2023*) and tumorigenesis (*McCleland et al., 2015*) as well as in osteoclastogenesis (*Yamada et al., 2022*) after tissue-specific *Cdk8* knockout. Although two CDK8/19 inhibitors were reported to have systemic toxicity (*Clarke et al., 2016*), this toxicity was subsequently found to be due to off-target effects of these inhibitors (*Chen et al., 2019*), and several CDK8/19 inhibitors have reached clinical trials (https://clinicaltrials.gov/ NCT03065010, NCT04021368, NCT05052255, NCT05300438). Several studies reported the existence of kinase-independent phenotypic activities for both CDK8 (*Menzl et al., 2019b*; *Kapoor et al., 2010*) and CDK19 (*Audetat et al., 2017*; *Steinparzer et al., 2019*), but the only known biochemical activity of both CDK8 and CDK19 that is kinase-independent is the stabilization of their binding partner CCNC (*Barette et al., 2001*; *Chen et al., 2023*). Targeted degradation or knockout of both CDK8 and CDK19 dramatically reduces the cellular levels of CCNC, whereas the knockout of either *Cdk8* or *Cdk19* alone has little effect on the CCNC levels (*Chen et al., 2023*). The effects of double knockout of both *Cdk8* and *Cdk19* on CCNC should, therefore, be considered in interpreting the effect of the double knockout.

We have now generated mice with a conditional knockout of the *Cdk8* gene on the constitutive *Cdk19* KO background (CDK8/19 inducible double knockout, iDKO). We have found that iDKO males were completely infertile and had an undersized and dedifferentiated reproductive system. In iDKO males, spermatogenesis was blocked after pachytene of meiosis I. iDKO showed down-regulation of key steroid pathway genes, such as *Cyp17a1*, *Star* and *Lcn2*, *Sc5d*, *Fads2*, and reduced production of testosterone. Expression of key meiotic genes was also deregulated in meiosis I spermatocytes,

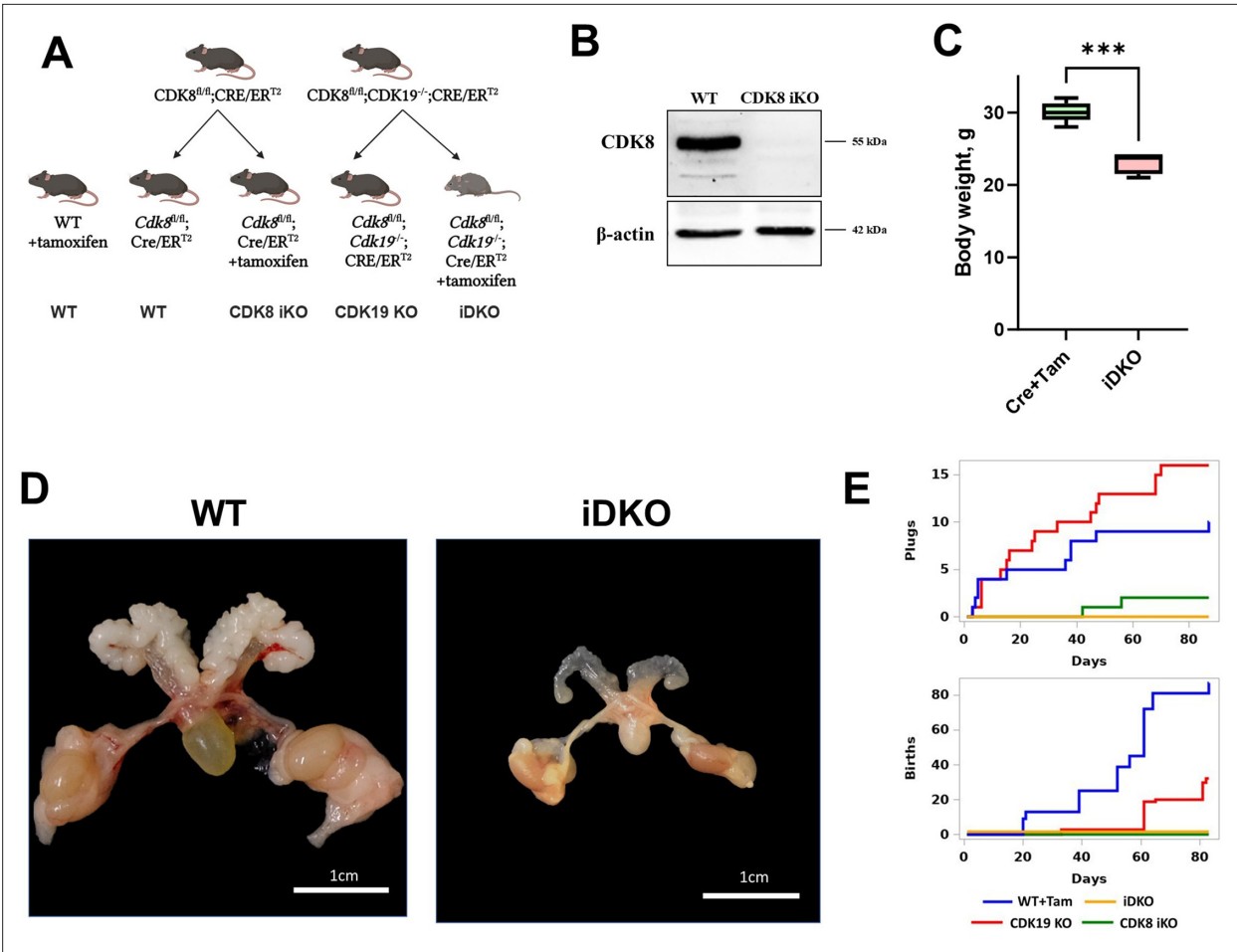

**Figure 1.** Changes in male urogenital system in *Cdk8/19* knockout mice. (**A**) Crossing of *Cdk8*^fl/fl^, *Cdk19*^-/-^ and Cre/ER^T2^ mice and formation of experimental (*Cdk8*^fl/fl^/*Cdk19*^-/-^/ ROSA26^CreERT2^+tamoxifen, *Cdk8*^fl/fl^/ROSA26^CreERT2^+tamoxifen, and *Cdk19*^-/-^) and control (*Cdk8*^fl/fl^/*Cdk19*^-/-^/ROSA26^CreERT2^ without tamoxifen and wild-type +tamoxifen) groups. (**B**) Confirmation of tamoxifen-induced CDK8 iKO in testes by Western blot. (**C**) After 2 mo of KO induction, iDKO mice had significantly lower body weight [nonparametric t-test, ***p<0.001, n=6]. (**D**) Male urogenital system atrophy in iDKO mice. (**E**) Sexual behavior and fertility of tamoxifen-treated control, single KOs, and iDKO male mice.

The online version of this article includes the following source data and figure supplement(s) for figure 1:

**Source data 1.** PDF file containing original western blots for *Figure 1B* indicating the relevant bands and treatments.

**Source data 2.** Original files for western blot analysis displayed in *Figure 1B*.

**Figure supplement 1.** Genotype and phenotype confirmation.

**Figure supplement 1—source data 1.** PDF file containing original gel for *Figure 1—figure supplement 1B and C* indicating the relevant bands and treatments.

**Figure supplement 1—source data 2.** Original files for gel analysis displayed in *Figure 1—figure supplement 1B and C*.

and the progression of meiosis was halted. These results indicated a key role of the CDK8/19/CCNC complex in the maintenance of male reproductive system.

## Results

### Spermatogenesis is blocked in CDK8/19 iDKO males

Previously, we have crossed *Cdk8*^fl/fl^ mice with ROSA26^CreERT2^ with tamoxifen-inducible Cre activity (*Figure 1A*; *Figure 1—figure supplement 1A*) and demonstrated effective KO in all tissues except for ovaries and uterus (*Ilchuk et al., 2022*). However, *Cdk8* inactivation in the male reproductive system (testes) was efficient at both genomic and protein levels (*Figure 1—figure supplement 1B*; *Figure 1B*). To investigate the effects of the double knockout of *Cdk8* and *Cdk19*, we crossed *Cdk8*^fl/^

$^{fl}$/ROSA26$^{CreERT2}$ mice with $Cdk19^{-/-}$ mice (**Zhang et al., 2022**) [MGI:5607862, $Cdk19^{em1(IMPC)J}$]. All the substrains were then maintained as homozygotes. We injected tamoxifen into 2-mo-old $Cdk8^{fl/fl}$/$Cdk19^{-/-}$/ROSA26$^{CreERT2}$ male mice 2 mo after tamoxifen injection. iDKO mice had significantly lower body weight than tamoxifen-injected control mice (**Figure 1C**). The most noticeable difference was the reduction of the male reproductive system - testes, epididymis, and prostate (**Figure 1D**). Changes observed in iDKO intestines resembled those detected in $Cdk8^{iIEC-KO}$/$Cdk19^{-/-}$ (**Dannappel et al., 2022**): the number of Paneth cells and goblet cells was significantly decreased (**Figure 1—figure supplement 1C and D**). Age-matched tamoxifen-treated ROSA26$^{CreERT2}$ mice served as a control.

Next, we decided to assess if the morphological changes in the reproductive system in the iDKO result in changes in their sexual behavior or the number of pups they fathered. As tamoxifen has an impact on the male reproductive system (**Willems et al., 2011**), the experiments were performed at least 6 wk post-injection and we used tamoxifen-treated C57BL/6 J mice as a control. Four groups were enrolled in the experiment: tamoxifen-treated C57BL/6 J (WT+Tam), tamoxifen-treated $Cdk8^{fl/fl}$ROSA-Cre/ER$^{T2}$ (CDK8 iKO), $Cdk8^{fl/fl}Cdk19^{-/-}$ROSA26$^{CreERT2}$ (CDK19 KO) and tamoxifen-treated $Cdk8^{fl/fl}Cdk19^{-/-}$ROSA26$^{CreERT2}$ (iDKO). Three males in each group were kept separately with two outbred CD1 females each for 3 mo. Copulative plugs and the number of pups were checked 5 d a week (**Figure 1E**). Tamoxifen-treated wild-type mice demonstrated normal fertility: 10 plugs were detected and 86 pups were born. The CDK19 KO group showed slightly reduced fertility: 32 pups along with an increased number of the plugs (16) probably caused by the lower rate of pregnancy onset. Surprisingly, only two plugs and no pups were observed in the CDK8 iKO group despite normal appearance and behavior of males. Neither plugs nor pups were detectable in the iDKO cohort (**Figure 1E**). These experiments showed that both CDK8 iKO and iDKO are infertile, and the cause of CDK8 iKO infertility is likely to be the lack of sexual activity.

We injected tamoxifen into 2-mo-old $Cdk8^{fl/fl}$/$Cdk19^{-/-}$/ROSA26$^{CreERT2}$ male mice. Age-matched tamoxifen-treated wild-type mice served as a control. To investigate iDKO effects we performed an autopsy with subsequent H&E staining, 8 wk post tamoxifen treatment (**Figure 2A**). The prostate, testes, and the epididymis were significantly smaller compared to single KOs and tamoxifen-treated wild-type mice (**Figure 2A**), the diameter of testicular tubules was reduced (**Figure 2B**).

The germinal epithelium of the tubules was presented by Sertoli cells with typical large light nuclei and some spermatogonia or spermatocytes. Cells at the postmeiotic stages of spermatogenesis were completely absent. Among the spermatocytes, there were only early prophase cells. In contrast, we detected an increased number of apoptotic cells (**Figure 2B**). Thus, in these mice the spermatogenesis progenitor cells were present but their differentiation and/or meiosis was blocked. In Sertoli cells, the vacuoles are the sign of the loss of contact between these cells and spermatogenic elements (**Figure 2A and B**). Leydig cells were significantly smaller and were almost depleted of secretory vacuoles suggesting the lack of hormonal activity. The epididymis contained empty tubules; the epithelium appeared undifferentiated, that is, neither typical borders between epithelial cells nor clear Golgi zone were visible. The prostate and the epididymis were also atrophic (**Figure 2A**). These changes were detectable from 2 wk after tamoxifen treatment and persisted for at least 6 m (**Figure 2C**). However, surprisingly they were not observed in CDK8 iKO (**Figure 2—figure supplement 1A**), which despite showing no microscopic changes, fathered no offspring and had no apparent sexual behavior (**Figure 1E**).

## CDK8/19 distribution in testes

Spermatogenesis failure in iDKO can be caused both by events in testes and extratesticular regulation. Internal events require expression of CDK8/19 at least in some testicular cells and an efficient KO of CDK8, which was not achieved in the female reproductive system (**Ilchuk et al., 2022**). Both CDK8 and CDK19 were present in testes and knockout was efficient after 7 d of daily treatment with 3 mg of tamoxifen (**Figure 2D**). We have also analyzed the levels of the CDK8/19 binding partner CCNC after single and double KOs in mouse embryonic fibroblasts (MEFs) (**Figure 2—figure supplement 1B**) and in mouse testes after iDKO (**Figure 2D**). In agreement with the previous observations in human cell lines (**Chen et al., 2023**), CCNC was present if either CDK8 or CDK19 were present, but undetectable in the iDKO embryonic fibroblasts, as well as in iDKO testes. As CDK8 and CDK19 functions are considered to significantly overlap (**Chen et al., 2023**), it is possible that in the absence of one of the proteins the level of the other will increase, obscuring a possible single KO phenotype.

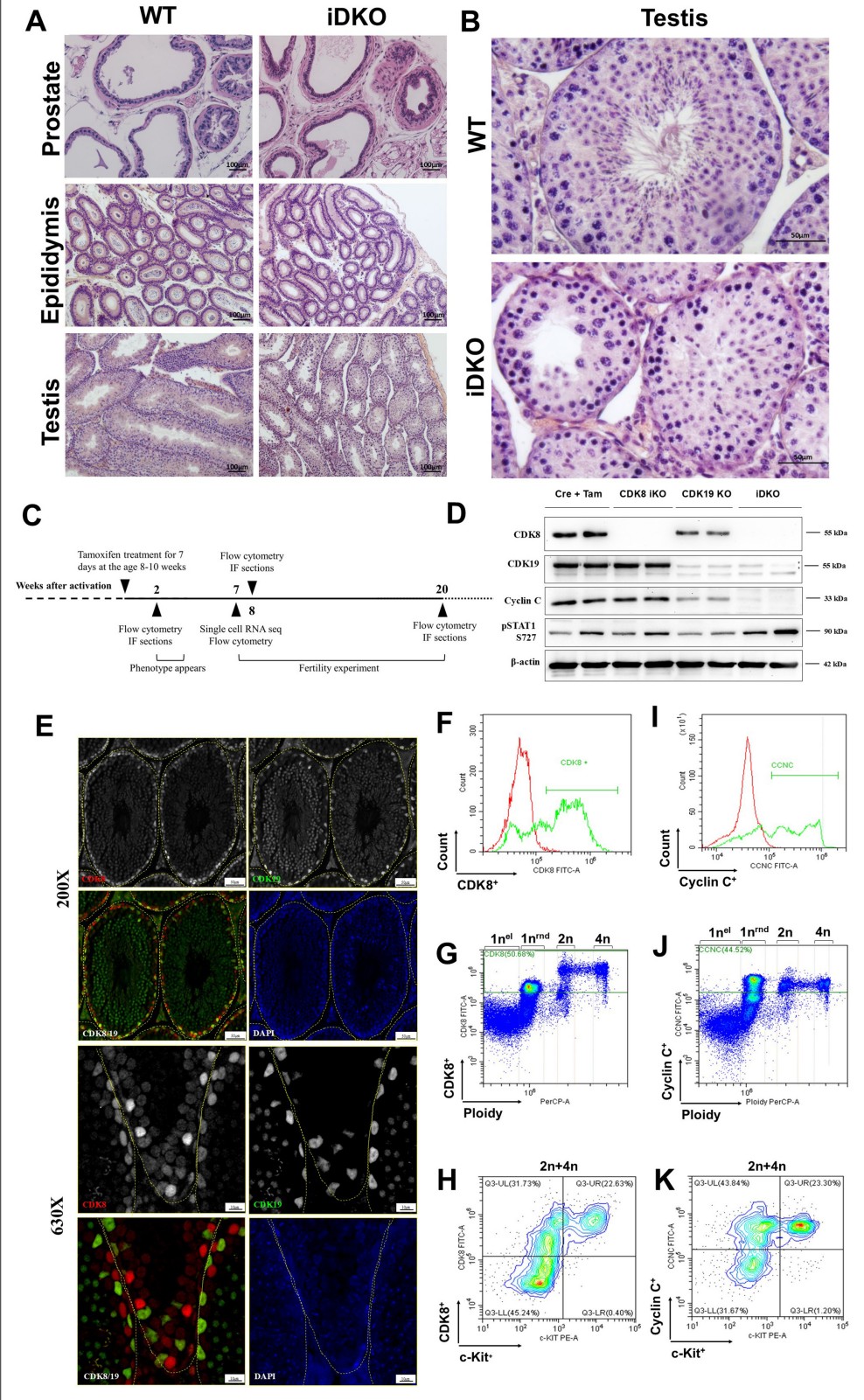

**Figure 2.** *Cdk8/19* knockout blocks spermatogenesis in mice. (**A**) H&E staining of prostate, epididymis, and testes of inducible double knockout (iDKO) mice and tamoxifen-treated control. 100 X magnification (**B**) H&E staining of wild-type (WT) and iDKO seminiferous tubules. 400 X magnification. (**C**) Time course of experiments. CDK8 iKO was activated by tamoxifen administration in 8–10 wk old males. Urogenital abnormalities became visible in

*Figure 2 continued on next page*

*Figure 2 continued*

2 wk. Spermatogenesis was analyzed by flow cytometry and immunofluorescence (IF) after 2, 8, and 20 wk since activation. Single-cell RNA sequencing was performed at 7 wk after KO. (**D**) Western blot analysis of Cre/ERT2 (Cre+Tam), single (CDK8 iKO and CDK19 KO), and double (iDKO) knockout testes, 2 mo after tamoxifen injections. CCNC protein is absent in iDKO, but not in the single KO in the testes. pSTAT1 727 is independent of CDK8/19 KO. Stars mark nonspecific staining by CDK19 antibodies. (**E**) CDK8 and CDK19 IHC staining of testes sections, 200 x (upper row) and 630 x magnification (bottom row) showing staining in various types of testicular cells. (**F–K**). Flow cytometry analysis of CDK8/CCNC expression in different testicular cell types. Figures F and I show major CDK8 (50.68%) and CCNC populations (44.52%), figures G and J show that 1 n (round, but not elongated spermatids), 2 n and 4 n cells can be CDK8 and CCNC positive, figures H and K indicate, not only cKit$^+$ cells among 2 n-4n can be CDK8 and CCNC positive.

The online version of this article includes the following source data and figure supplement(s) for figure 2:

**Source data 1.** PDF file containing original western blots for *Figure 2D* indicating the relevant bands and treatments.

**Source data 2.** Original files for western blot analysis displayed in *Figure 2D*.

**Figure supplement 1.** CDK8 and CDK19 are mutually compensatory for cyclin C stabilization and male urogenital system phenotype.

**Figure supplement 1—source data 1.** PDF file containing original western blots for *Figure 2—figure supplement 1B* indicating the relevant bands and treatments.

**Figure supplement 1—source data 2.** Original files for western blot analysis displayed in *Figure 2—figure supplement 1B*.

**Figure supplement 2.** CDK19 antibody specificity.

Indeed, we observed such a compensation for MEFs, but not for testes (*Figure 2D*; *Figure 2—figure supplement 1B*), indicating tissue specificity of this mechanism. Decrease in the phosphorylation of STAT1 serine 727 is often used as CDK8/19 inhibition marker (*Rzymski et al., 2017*) despite the fact that different kinases can phosphorylate it (*Chen et al., 2019*). However, pSTAT1 727 level was not decreased in single or double KOs (*Figure 2D*) indicating that pSTAT1 727 is not a suitable marker of CDK8/19 kinase activity.

Different testicular cell types contribute to the successful spermatogenesis and little is known about CDK8/19 expression in each type. *McCleland et al., 2015* performed IHC on different tissues and found CDK8 only in spermatogonia of all testicular cell types. Our IHC analysis showed that CDK8 and CDK19 are present in several types of testicular cells, including Leydig cells, and cells inside the tubule. Interestingly, CDK8 and CDK19 appear to be expressed in different types of cells in the seminiferous tubules, with CDK8 mostly expressed in the periphery of the tubule and in lower levels in the center of the tubule, while CDK19 was mostly expressed in the center of the tubule (*Figure 2*; *Figure 2—figure supplement 2*). We also used flow cytometry to identify cell types expressing CDK8/19. To detect cells with the expression of at least one kinase we used anti-CCNC antibodies. In wild-type male mice, CDK8 (*Figure 2F–H*) and CCNC (*Figure 2I–K*) positive cells were detected among 1 n (round, but not elongated spermatids), 2 n and 4 n cells, indicating their role at all spermatogenesis stages. In agreement with (*McCleland et al., 2015*) report, all c-Kit$^+$ cells were positive for CCNC and CDK8, but other 2 n/4 n c-Kit negative cells were also positive for both proteins (*Figure 2H and K*).

## Spermatogenic cells of iDKO mice are unable to advance through meiosis I prophase

To obtain a quantitative spermatogenesis pattern, we performed cell cycle analysis by flow cytometry with propidium iodide staining. We examined wild-type, single KOs, and iDKO animals sacrificed 2 mo after tamoxifen treatment. Cell cycle analysis revealed striking differences in iDKO mice, with a disappearance of elongated spermatids, almost all round spermatids, and massive cell death (*Figure 3A and B*). All other groups had the same normal cell distribution (*Figure 3B*). Noticeably, the 4 n population and cells in the S-phase were not affected by iDKO, indicating that iDKO spermatogonia successfully entered meiosis but could not produce haploid spermatids. This ongoing 'meiotic catastrophe' led to an almost full depopulation of the testes in iDKO mice (*Figure 3C*). Due to fast transition from meiosis I to meiosis II, it is difficult to detect secondary spermatocytes by

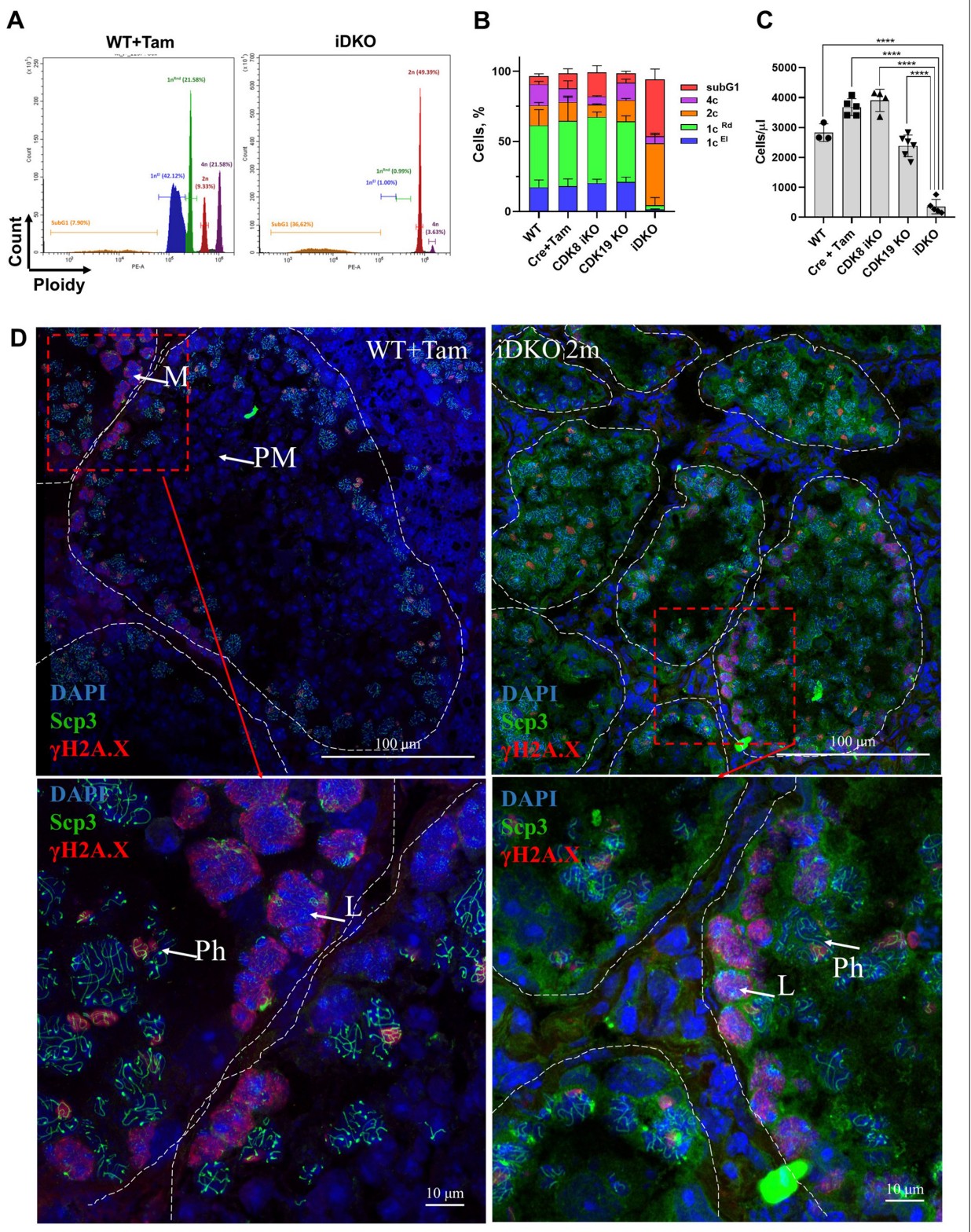

**Figure 3.** Absence of postmeiotic 1 n cells in inducible double knockout (iDKO) 2 mo after KO induction. (**A**) Distinctive histograms of wild-type (left) and iDKO (right) mice. Violet - 4 n population, red - 2 n population, green - round spermatids, blue - elongated spermatids, orange - apoptotic subG1 cells. (**B**) Quantitative distribution of testicular cells between these groups. Wild-type, with and without tamoxifen, as control groups, have similar distribution to that of CDK8 and CDK19 single KO. iDKO testes have a greatly reduced number of round spermatids and no elongated spermatids [repeated measurements two-way ANOVA, mean ± SD, n=5]. (**C**) Overall cellularity is significantly reduced only in the iDKO testes [nonparametric t-test,

*Figure 3 continued on next page*

*Figure 3 continued*

n > 3, ***p<0.001, ****p<0.0001]. (**D**) IF staining of the control and iDKO testes frozen sections. Nuclei are stained by DAPI (blue pseudocolor), SYCP3 is depicted as green, γH2A.X - as red. All stages of spermatogenesis are visible in control testes, while pachytene is the last detected stage in iDKO. Confocal microscopy, magnification 600 X. M - meiotic entry spermatocytes; L - leptotene; Ph - pachytene; PM - post-meiotic stages.

flow cytometry. To specify the stage of CDK8/19-dependent spermatogenesis failure, we performed phospho-γH2A.X and SYCP3 immunofluorescent staining of seminiferous tubules. Phospho-γH2A.X is a DNA damage marker which marks double-strand breaks in leptotene, zygotene, and sex chromosomes during the pachytene. SYCP3 is a component of the synaptonemal complex, expressed during meiosis I prophase, which forms distinct patterns in the late zygotene/pachytene. In the wild-type tubules, all the meiotic stages were visible (*Figure 3D*). At the same time, almost all meiotic cells in iDKO were blocked in the pachytene (*Figure 3D*), indicating that iDKO cells enter meiosis but cannot traverse through meiosis I prophase and subsequently undergo cell death.

## Single-cell RNA sequencing

To investigate molecular mechanisms of CDK8/19 mediated alterations in meiosis, we performed single-cell RNA sequencing (scRNAseq) of the testes. We analyzed cell suspensions from testes of two tamoxifen-treated wild-type C 57BL/6 J and two iDKO animals whose phenotype had previously been confirmed by cell cycle analysis (*Figure 4—figure supplement 1*).

The analysis of cell type composition confirmed the reduced number of post-pachytene spermatocytes and almost complete absence of spermatids, while meiotic entry, leptotene, and zygotene cell numbers remained unaffected (*Figure 4A and B*). At the same time, the percentage of undifferentiated spermatogonial stem cells in iDKO animals was unchanged, nor was the proportion of Leydig cells altered. Sertoli cells became the most abundant cell type in iDKO mice with reduced testes cellularity, however their absolute number didn't change significantly.

## Differential gene expression

Differentiation of sperm lineage is a tightly controlled process, where changes in functioning in a certain type of cells can affect the viability and differentiation of other cells. To understand the underlying mechanism of meiotic cell death in iDKOs, we compared gene expression in each type of cell clusters present both in wild-type and knockout animals.

## Leydig cells

Leydig cells are the primary source of testosterone in the testes, a hormone required for spermatogenesis. scRNAseq analysis showed significant transcriptomic changes in Leydig cells. Among 126 genes that were differentially expressed ($|\log2FC|>0.4$; p<0.01) in WT *vs* iDKO Leydig cells, 30 were down-regulated, however, among 14 strongly affected genes with $|\log_2FC|>1$ 11 were down-regulated. Strikingly, 9 of 14 strongly downregulated genes were associated with lipid (specifically, steroid) metabolism (*Supplementary file 1*), and 3 of these 9 genes were linked to male infertility (*Aherrahrou et al., 2020*; *Stoffel et al., 2008*; *Song, 2007*).

According to the GO analysis (*Figure 4C*; *Supplementary file 2*), the most prominent changes were the downregulation of lipid metabolism and the steroid hormone biosynthesis. The most significantly down-regulated gene was *Cyp17a1*, showing a 10-fold decrease in expression (*Figure 4D*). The *Cyp17a1* gene in male mice is mainly expressed in Leydig cells (*Missaghian et al., 2009*). The product of this gene catalyzes the key reaction of the steroidogenic pathway. Mice with constitutive *Cyp17a1* KO are phenotypically female (*Aherrahrou et al., 2020*). Downregulated *Lcn2* (*Figure 4D*) specifically expressed in Leydig cells codes for a protein of the lipocalin family that transports small hydrophobic molecules, including steroids, and is affected by fertility manipulations (*Kang et al., 2017*; *Yanai et al., 2021*). Another downregulated gene, *Fads2* (*Figure 4D*), is a key enzyme for biosynthesis of highly unsaturated fatty acids. Its KO causes infertility in males by arresting the spermatogonial cycle at the stage of the round spermatids (*Stroud et al., 2009*). The *Sult1e1* (*Figure 4D*) gene encodes sulfotransferase which inactivates estrogens and its deficiency leads to hyperplasia of Leydig cells and atrophy of seminiferous tubules (*Song, 2007*). The *Star*, *Sc5d*, *Pld3*, and *Apoc1* (*Figure 3D*) genes, also involved in lipid metabolism, were down-regulated more than twofold. Interestingly, the *Hsd3b6* (*Figure 4D*) is a surprisingly upregulated gene associated with steroid metabolism in Leydig cells.

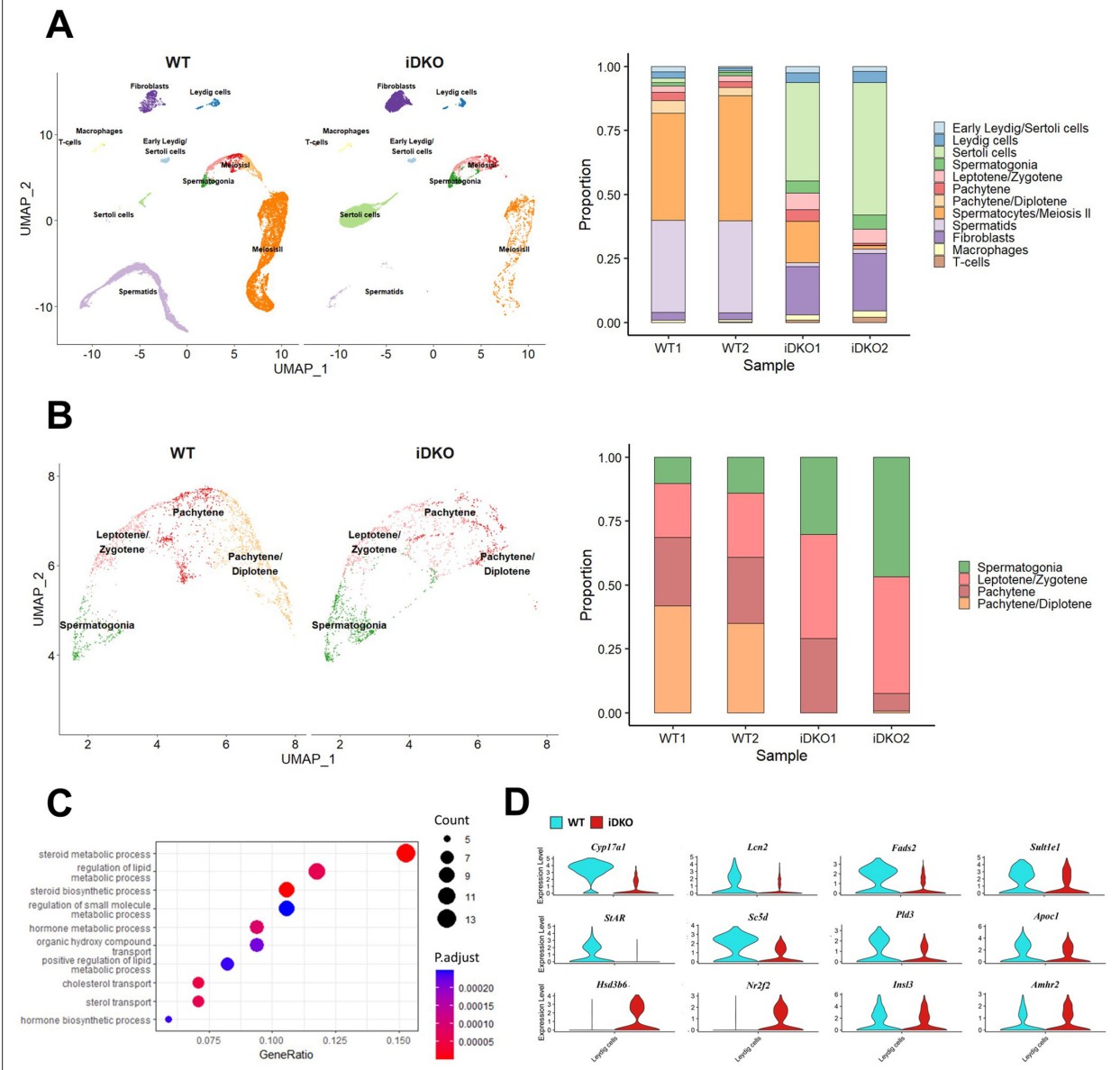

**Figure 4.** Single-cell RNA sequencing reveals loss of spermatids due to steroidogenesis failure. Raw data single-cell RNA sequencing (scRNA) sequencing are available in the SRA (SRP470231). (**A**) UMAP projection and relative cell numbers for all testicular cell types in control and iDKO samples. Number of secondary spermatocytes is significantly decreased and spermatids are almost absent in iDKO samples. (**B**) UMAP projection and relative cell numbers for spermatogonia and primary spermatocytes. Post-pachytene spermatocytes are severely depleted in iDKO samples. (**C**) GO Biological Processes pathways enriched among the Leydig cells differentially expressed genes (DEGs). Lipid metabolism and steroid biosynthesis are severely perturbed. (**D**) Violin plots for key Leydig cells genes.

The online version of this article includes the following figure supplement(s) for figure 4:

**Figure supplement 1.** Flow cytometry analysis of propidium iodide stained cell fractions from testes of 2 wild-type (WT) and 2 double knockout (DKO) testes later subjected to single-cell RNA sequencing (scRNAseq).

Not only testosterone production was decreased in double knockouts. Expression of *Insl3* gene, coding for an important peptide hormone secreted by Leydig cells (***Esteban-Lopez and Agoulnik, 2020***), was also reduced (***Supplementary file 3***).

Among up-regulated genes, there were several transcription factors governing steroidogenesis in Leydig cells (***Ye et al., 2017***; ***de Mattos et al., 2022***). One of them, *Nr2f2* (***Figure 4D***) encodes COUP-TFII which regulates genes immediately involved in steroidogenesis (*Cyp17a1*, *Hsd3b1*, *Cyp11a1*, and *Akr1c14*) alongside several other Leydig-specific genes (*Insl3*, *Amhr2*) (***Figure 3D***). Another gene,

*Cebpb*, encodes TF C/EBPβ which activates *Star* transcription and stimulates expression of *Nr4a1* (also known as *Nur77*) which is itself a key regulator of steroidogenesis in Leydig cells.

The *Kit* gene (down-regulated), encoding a tyrosine kinase receptor, and *Kitl* gene (up-regulated), encoding its ligand, were also present among the differentially expressed genes. Both of these genes play an important role in spermatogenesis (knockouts and mutants are infertile), Leydig cell maturation, and steroidogenesis (*Liu et al., 2017*; *Ye et al., 2017*).

We hypothesize that the observed upregulation of several steroidogenesis regulators may be a part of a compensation feedback loop in response to the decreased testosterone level.

## Sertoli cells

ScRNAseq (*Figure 4A*) shows that Sertoli cell fraction is greatly increased in iDKO specimens, however, this is mostly due to the reduction of the total germ cell number. At the same time, analysis of cell cycle-specific genes revealed that in iDKO, Sertoli cells can re-enter the cell cycle. In particular, genes involved in G0-S transition and G2-M transition were upregulated (*Figure 5A*).

According to downregulation of the steroid hormone biosynthesis in Leydig cells, we expected to see in Sertoli cells patterns similar to those in mice with Sertoli cell-specific KOs of androgen receptor (AR) – SCARKO (Sertoli Cells Androgen Receptor KnockOut) (*Larose et al., 2020*; *De Gendt et al., 2014*) and luteinizing hormone (LH) receptor – LURKO (*Griffin et al., 2010*). However, it must be noted that these mice have constitutive receptor KOs and do not fully develop their reproductive system, whereas our KO model mice develop normally and become fertile before the knockout induction.

Among DEGs, we identified a number of upregulated and downregulated genes consistent with their changes in SCARKO (*Defb45*, *Myl6*, *Tmsb4x*, *Espn*, and *Ldhb* among top DEGs). At the same time, we compared our DEG set with those published earlier (*Larose et al., 2020*; *De Gendt et al., 2014*) and observed little overlap between our data and published results as well as between the two published papers themselves (*Figure 5—figure supplement 1*).

However, a much better agreement was found between our data and the study by deGendt et al. (*De Gendt et al., 2014*) at the level of GO Molecular Function and Cellular Compartment categories.

According to the GO Cellular Compartment (*Supplementary file 2*), both up- and down-regulated genes are associated with cytoskeleton (*Vim*, *Actb*, *Actg1*) (*Show et al., 2003*), apical-basal axis (*Cdc42*) (*Heinrich et al., 2021*), cellular membrane (*Atp1a1*, *Gja1*) (*Rajamanickam et al., 2017*; *Sridharan et al., 2007*), and intercellular contacts (*Espn*, *Anxa2*, *S100a10*) (*Willems et al., 2010*; *Chojnacka et al., 2017*). These closely related structures play an important role in Sertoli cell functioning: formation of the blood-testis barrier and intercommunication with developing germ cells (*Figure 5B*; *Wong and Cheng, 2009*). In agreement with this, immunostaining also revealed perturbed vimentin polymerization (*Figure 5C*). Therefore, our findings on differential gene expression in Sertoli cells are in agreement with data obtained in mice with impaired AR signaling. Among the top enriched biological processes several up-regulated stress response pathways were also found, indicating deterioration of the Sertoli cells in the absence of testosterone (*Figure 5D*).

This molecular evidence of Sertoli cell spatial organization disturbance and disruption of cell contacts is in good agreement with the patterns observed during histological examination.

## Germ cells

As primary spermatocytes of double knockout mice stop their differentiation in pachytene, we compared gene expression in two groups of germ cells: undifferentiated spermatogonia and meiosis-entry (leptotene-zygotene-pachytene) spermatocytes. 1107 DEGs were identified for undifferentiated cells, 1650 for spermatocytes, and 874 of these genes were common. Almost all the DEGs were up-regulated (99.2%) (*Figure 5—figure supplement 2*). This overlap may be explained by the fact that undifferentiated and differentiated spermatogonia represent a continuous spectrum with gradually changing transcription patterns. Unique genes for the Early Spermatogonia cluster were attributed by GO as genes related to the RNA processing and biosynthesis and ribosome biogenesis, whereas no stem cell-specific pathways were found. We conclude that *Cdk8/19* knockout and disruption of steroid biosynthesis and testosterone production have no significant impact specifically on the early spermatogonia.

Most cells in knockout testes do not progress to stages after pachytene. Accordingly, apoptosis-related pathways, stress response pathways, and the autophagy pathway were upregulated

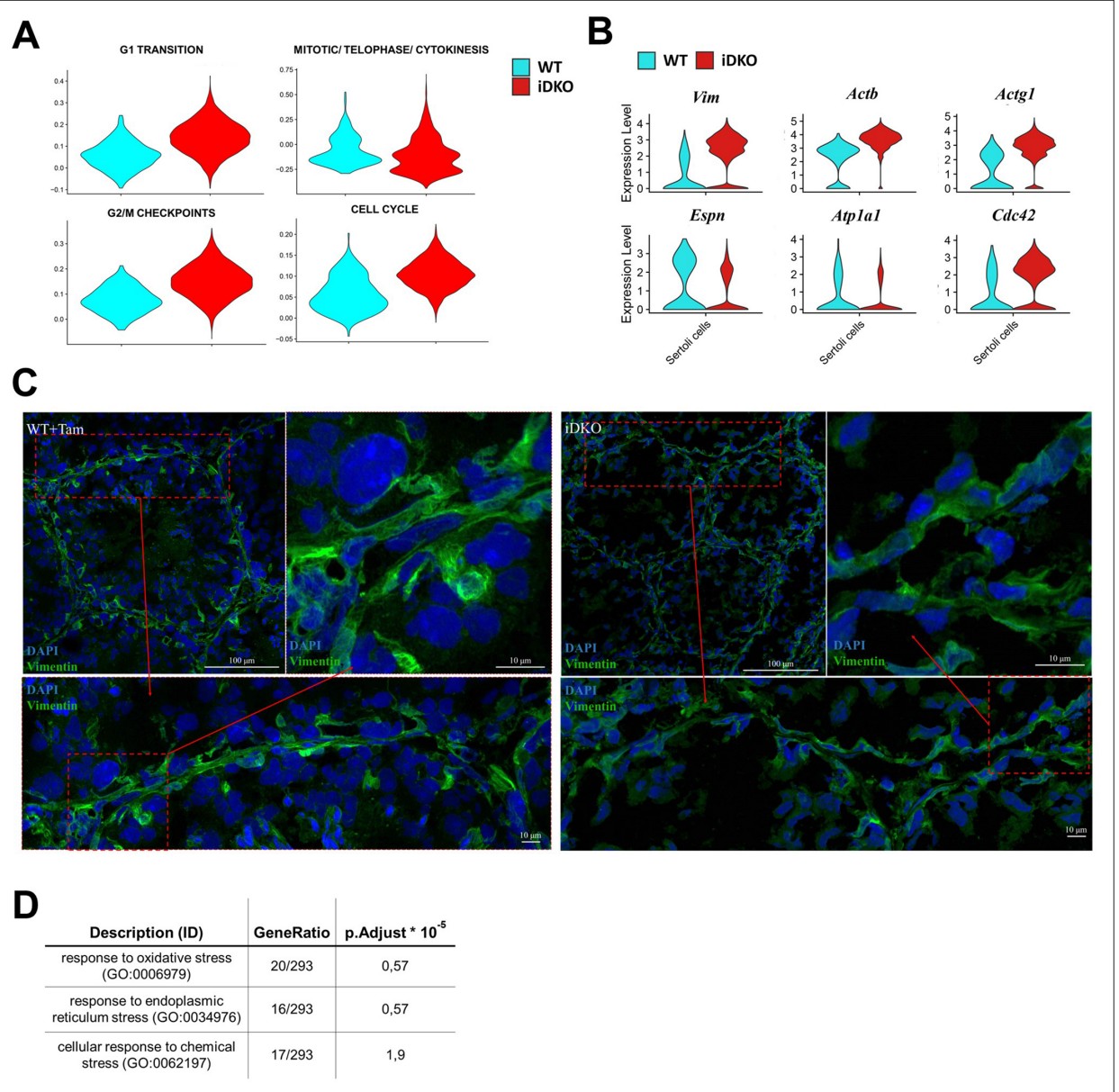

**Figure 5.** Inducible double knockout (iDKO) Sertoli cells re-enter the cell cycle and lose characteristic cytoskeleton organization. (**A**) Violin plots for Reactome cell cycle gene sets indicate that Sertoli cells in iDKO lose terminal differentiation and re-enter cell cycle. Percentage of cells in G1-S and G2-M transitions are increased in iDKOs. (**B**) Violin plots for key cytoskeleton and intercellular contacts related to differentially expressed genes (DEGs). (**C**) Immunofluorescence (IF) staining for vimentin demonstrates the blood-testis barrier (BTB) integrity disruption and a loss of characteristic striation cytoskeleton patterns in iDKOs. Magnification 600 X. (**D**) Enrichment of GO stress pathways in Sertoli cells indicates their dysfunction in iDKOs.

The online version of this article includes the following figure supplement(s) for figure 5:

**Figure supplement 1.** Venn diagram of SCARCO, SCARIBO и iDKO mice differentially expressed genes (DEGs).

**Figure supplement 2.** Volcano plots of differentially expressed genes (DEGs) of spermatogonia and spermatocytes clusters (**A**), Sertoli (**B**), and Leydig cells (**C**).

**Figure supplement 3.** GO terms for spermatogonia depicted as a treemap.

(*Supplementary file 2*; *Figure 5—figure supplement 3*). More specifically, a set of meiosis-specific genes was also up-regulated. These genes were related to chromosome pairing, DNA recombination, and histone modification, which are known to be involved in meiosis. We hypothesize that this effect might be caused by compensatory mechanisms triggered by inability to proceed through meiosis.

## Validation of single-cell RNA data

In control testes, CYP17A1 protein was localized outside of seminiferous tubules, in Leydig cells. In iDKO animals, CYP17A1 staining was completely absent (*Figure 6A*), despite Leydig cells being present in histological sections (*Figure 2A and B*). Western blot analysis of all genotypes further confirmed this finding (*Figure 6B*).

CYP17A1 catalyzes several key reactions leading to transformation of pregnenolone into testosterone. We measured the concentration of the key androgen hormone -- testosterone in the blood of wild-type mice, single KOs, and iDKO. Only iDKO showed significantly lower levels of testosterone, consistent with only this genotype presenting abnormal spermatogenesis (*Figure 6C*).

Changes in steroid synthesis in Leydig cells can be caused by several mechanisms. Specifically, CDK8/19/CCNC can regulate CYP17A1 expression in Leydig cells themselves, or CDK8/19/CCNC can be involved in regulation of brain hormonal signals, which control testosterone production. To test the latter hypothesis, we measured the concentration of luteinizing hormone in blood serum. The results showed no difference in LH level through all the genotypes (*Figure 6D*). Therefore, it is more plausible that CDK8/19/CCNC acts directly in Leydig cells to regulate steroidogenesis.

CDK8 was also shown to positively regulate lipid metabolism in *D. melanogaster* (*Tang et al., 2018*; *Li et al., 2022*). In this case, a lack of steroidogenesis could be explained by the lack of its precursor cholesterol. We detected OilRed-positive stained cells in the intertubular space in good agreement with the Leydig cells localization. Moreover, iDKO staining was more intense than in wild-type mice, indicating an accumulation of lipids, which are not converted to testosterone (*Figure 6— figure supplement 1*).

## Spermatogenesis in iDKO mice slightly recovers with time

To address the question of the persistence of spermatogenesis failure, we performed histological and flow cytometry analysis of mice, 5 mo after tamoxifen treatment. A certain percentage of 1 n cells (primarily round spermatids) have reappeared 3–5 mo after tamoxifen treatment (*Figure 7A*), nevertheless the total number of cells and 1 n cells remained low compared to control (*Figure 7B*). H&E staining of prostate and epididymis also revealed slight alleviation of the phenotype, but no mature sperm (*Figure 7C*). Immunofluorescent staining of seminiferous tubules confirmed this finding, demonstrating solitary tubules with postmeiotic cells (*Figure 7D*). However, the levels of CYP17A1 were not restored in Leydig cells (*Figure 7E*).

## Pharmacological inhibition of CDK8/19 does not affect spermatogenesis

Recent findings have established that CDK8/19 may have kinase-independent functions (*Steinparzer et al., 2019*), including stabilization of CCNC in a kinase-independent manner (*Chen et al., 2023*). Depletion of CDK8/19 in testes of iDKO mice leads to the CCNC degradation (*Figure 2D*). To examine if the observed phenotype is related to the kinase activity of CDK8/19 or is kinase-independent, we treated WT mice with a pharmacological CDK8/19 inhibitor – SNX631-6 (*Li et al., 2024*). 11 wk-old male C57BL/6 J mice were treated with SNX631-6 medicated chow (500 ppm, 40–60 mg/kg/d dosage, on average) or control diet for 3 wk. The same SNX631-6 dosing regimen was highly efficacious in suppressing the castration-resistant prostate cancer growth in male mice (*Li et al., 2024*). No abnormal clinical observations were identified in the treated mice through the whole treatment period. Testes were weighed and fixed in 10% formalin for H&E staining. There were no significant differences between the control and treatment groups in histology analysis or in testes weights (*Figure 8A and B*), in contrast to the iDKO testes. The drug concentration in the testes as measured by LCMS/MS was lower than in blood (*Figure 8C*), suggesting the effects of the blood-testis barrier, but this concentration (50 ng/mL) was still ~10 times higher than the drug's $IC_{50}$ in the cell-based assay (~5 ng/mL).

The effects of iDKO and CDK8/19 kinase inhibition were compared in ex vivo primary Leydig cell culture from WT mice and *Cdk8*$^{fl/fl}$*Cdk19*$^{-/-}$ROSA26$^{CreERT2}$ mice. We treated the former with CDK8/19 inhibitor Senexin B and the latter with hydroxytamoxifen to induce *Cdk8*$^{-/-}$ in vitro. Then, we compared expression of *Cyp17a1*, *Star,* and *Fads* genes that were downregulated according to the scRNAseq data in treated and untreated cells (*Figure 8D*). *Star* and *Fads* were downregulated in OH-Tam treated cells, whereas basal *Cyp17a1* expression level was too low to make any conclusion. At the same time,

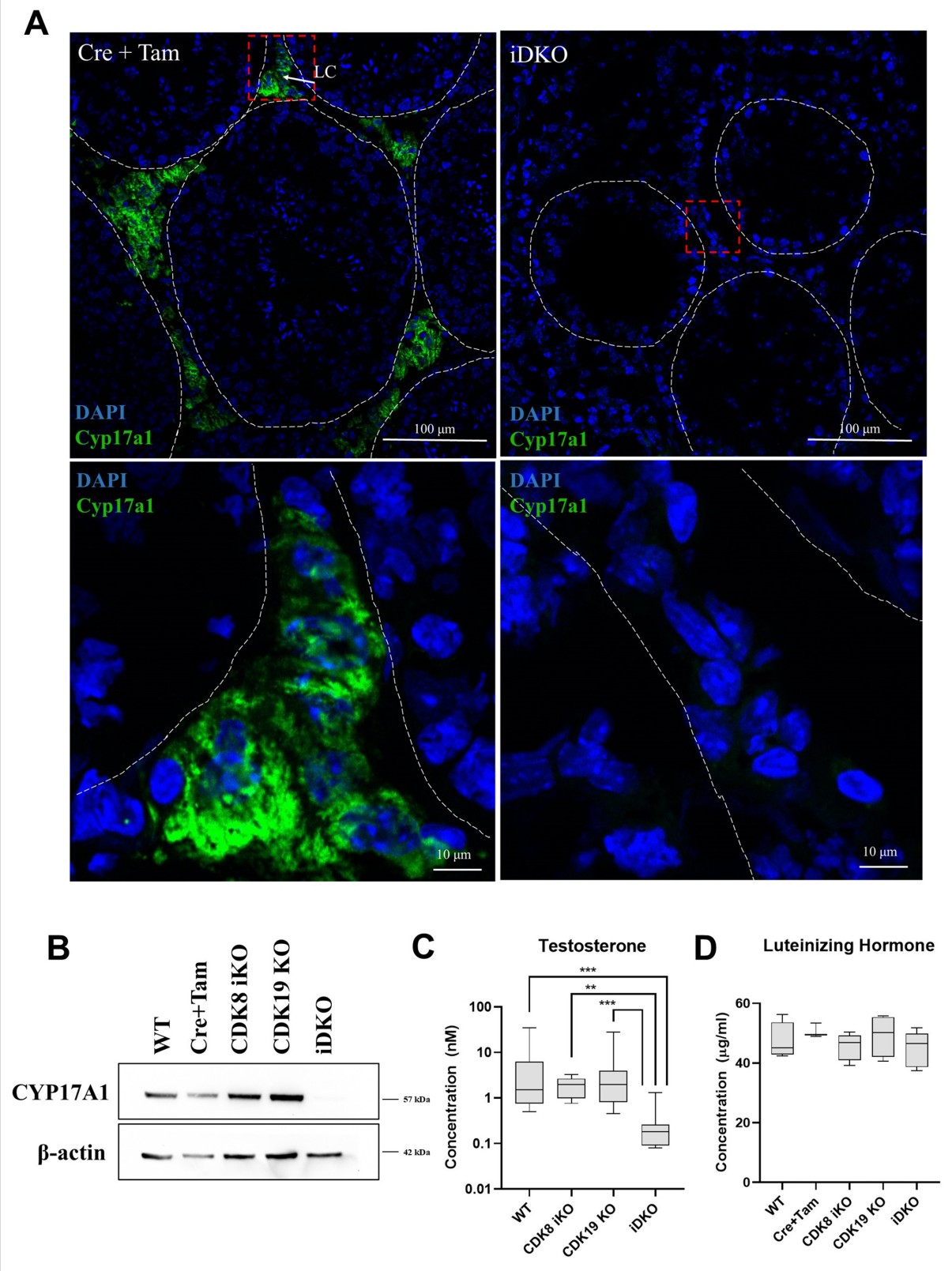

**Figure 6.** Confirmation of single-cell RNA sequencing (scRNA) sequencing data by other methods. (**A**) Immunofluorescence (IF) staining for CYP17A1 of testes frozen sections, magnification 600 X. CYP17A1 is visualized in extratubular space in Leydig cells in control mice and is completely absent in inducible double knockout (iDKOs). (**B**) Western blot for CYP17A1 confirms disappearance of the protein in iDKOs, but not in other genotypes

*Figure 6 continued on next page*

*Figure 6 continued*

[nonparametric t-test, n=5, **p<0.01, ***p<0.001]. (**C**) Testosterone blood level is decreased only in iDKO mice. (**D**) Luteinizing hormone production is not impaired by CDK8 iKO and CDK19 KO or iDKO [nonparametric t-test, n=5].

The online version of this article includes the following source data and figure supplement(s) for figure 6:

**Source data 1.** PDF file containing original western blots for *Figure 6B* indicating the relevant bands and treatments.

**Source data 2.** Original files for western blot analysis displayed in *Figure 6B*.

**Figure supplement 1.** OilRed-stained frozen sections of testes of Cre+Tam and inducible double knockout (iDKO) mice 2 mo of tamoxifen injection.

CDK8/19 kinase inhibitor did not affect expression levels of *Star* and *Fads*, suggesting that the effects of the iDKO were kinase-independent.

## Discussion

The Mediator complex is a key part of transcriptional regulation through signaling pathways. The enzymatic components of the Mediator-associated CKM module – CDK8 and its paralog CDK19, have emerged as co-regulators in several signal-activated transcriptional pathways, such as Wnt, TGF-beta, androgen and estrogen signaling, STATs, response to serum growth factors, and others (reviewed in *Menzl et al., 2019a*). Despite that, the knockout of *Cdk8* is only critical in embryogenesis (*Westerling et al., 2007*), with no phenotype in adult organisms (*McCleland et al., 2015*), while *CDK19* is wholly dispensable. The same tendency is observed for the knockout of MED subunits and other CKM proteins: constitutive KOs are often embryonically lethal (except for MED12L, a paralog of MED12), while tissue-specific knockouts of these genes in adult animals rarely have a severe phenotype (*Ilchuk et al., 2023*).

Hypothetically, CDK8 and CDK19 can compensate for each other in KO animals and mitigate the phenotype in single KOs. To confirm this, we produced the first ubiquitous knockout of *Cdk8/19* in 2-mo-old animals. In addition to the specific phenotype described here, the analysis of tissues from the knockout animals provided the first in vivo confirmation of two key molecular observations on the general CDK8/19 activities that were previously described only in vitro and that are not yet widely understood as potential caveats in CDK8/19 studies. The first observation (*Chen et al., 2023*) is that the knockout of CDK8 and CDK19 leads to the loss of CCNC, in contrast to CDK8/19 kinase inhibition that actually increases the protein levels of CCNC. In the present study, we found that CCNC is depleted both in MEF and in the testes of iDKO mice. This finding underscores that CDK8/19 kinase inhibition may have very different phenotypic consequences from the knockout of these genes. The second finding was that STAT1 phosphorylation at serine 727, a widely used pharmacodynamic marker of CDK8/19 kinase activity, shows variable response to CDK8/19 inhibition, which depends on the cell type and the duration of CDK8/19 inhibitor treatment, and that it can be induced by different signals in a CDK8/19-independent manner (*Chen et al., 2019*). Here, we found that the serine 727 phosphorylated STAT1 was undiminished (and possibly even increased) by *Cdk8/19* knockout in the testes of iDKO mice (*Figure 2D*). Hence, a reliance on STAT1 phosphorylation as a pharmacodynamic marker of CDK8/19 activity can be misleading.

Previously described animals with conditional knockout of *Cdk8* in the intestine on *Cdk19⁻/⁻* background presented with a mild phenotype, moderately affecting the number of specialized secreting cells (*Dannappel et al., 2022*). This phenotype was confirmed by histological observations in our model. The most prominent phenotype in our iDKO mice, evidenced at the physiological, morphological, and molecular levels, was found in the testes, matching the finding that the testes express the highest levels of CDK8 and CDK19 RNA among all the normal human tissues (*Li et al., 2024*). In particular, we observed the lack of haploid spermatids and mature spermatozoa (*Figure 2A–B*), with primary spermatocytes blocked in pachytene of meiosis I (*Figure 3*). CDK8 and CDK19 were expressed in different types of testicular cells, putatively identified as spermatogonia (high expression), Leydig cells, and Sertoli cells, and spermatocytes (weak expression) for CDK8, and Sertoli cells and spermatocytes (high expression) for CDK19. CDK8 and CDK19 were co-localized in some cells, although a substantial part of individual cells with high expression of CDK8 had weaker CDK19 staining and vice versa or expressed just one paralog (*Figure 2E*). We hypothesize that the specific CDK8/19 ratio can depend not only on the distinct cell type but also on the stage of the distinct

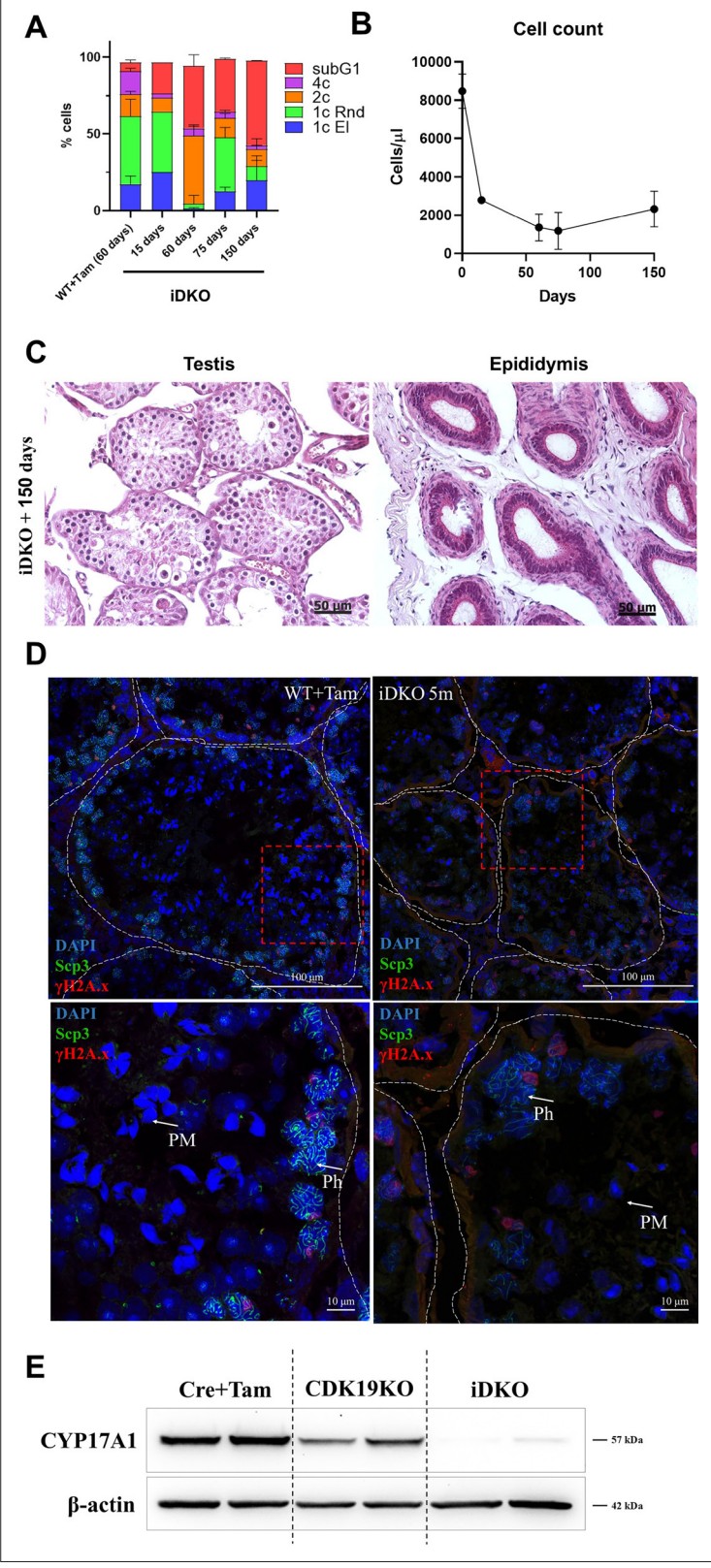

**Figure 7.** Limited recovery of spermatogenesis, 5 mo after inducible double knockout (iDKO) induction. (**A**) Round and elongated spermatids become detectable by flow cytometry 5 mo after iDKO [repeated measurements two-way ANOVA, mean ± SD, n=3]. (**B**) The overall testes cellularity is only slightly increased [nonlinear regression, mean ± SD, n=3]. (**C**) The postmeiotic cells become visible with H&E staining of the tubules, however, epididymal

*Figure 7 continued on next page*

*Figure 7 continued*

ducts remain empty. (**D**) Post-pachytene and post-meiotic (PM) cells became visible on the SYCP3 + γH2A.X-stained frozen sections, magnification 600 X. (**E**) CYP17A1 level remains at the background level, 5 mo after KO induction.

The online version of this article includes the following source data for figure 7:

**Source data 1.** PDF file containing original western blots for *Figure 7E* indicating the relevant bands and treatments.

**Source data 2.** Original files for western blot analysis displayed in *Figure 7E*.

tubule for the Leydig and Sertoli cells and on the precise meiotic stage for germ cells. Flow cytometry analysis also showed high levels of CDK8 and CCNC in 2 n, 4 n, and in some haploid round spermatids (but not elongated spermatids) (*Figure 2F-G, I-J*). We also confirmed previous reports of *McCleland et al., 2015* identifying CDK8 expression in c-Kit +spermatogonia cells, but also showing expression of CDK8 and CCNC in other 2 n and 4 n c-Kit- cells (*Figure 2H and K*). Surprisingly, CDK8 iKO males previously described as asymptomatic revealed their infertility due to the lack of sexual behavior, despite the absence of morphological changes in the urogenital system (*Figure 1E*, *Figure 1—figure supplement 1A*). As our results imply that fertility problems and spermatogenesis failure are caused by changes in the gene expression in the Leydig cells, it is important to mention that CDK8 is expressed in Leydig cells at a relatively high level, whereas CDK19 is also expressed but

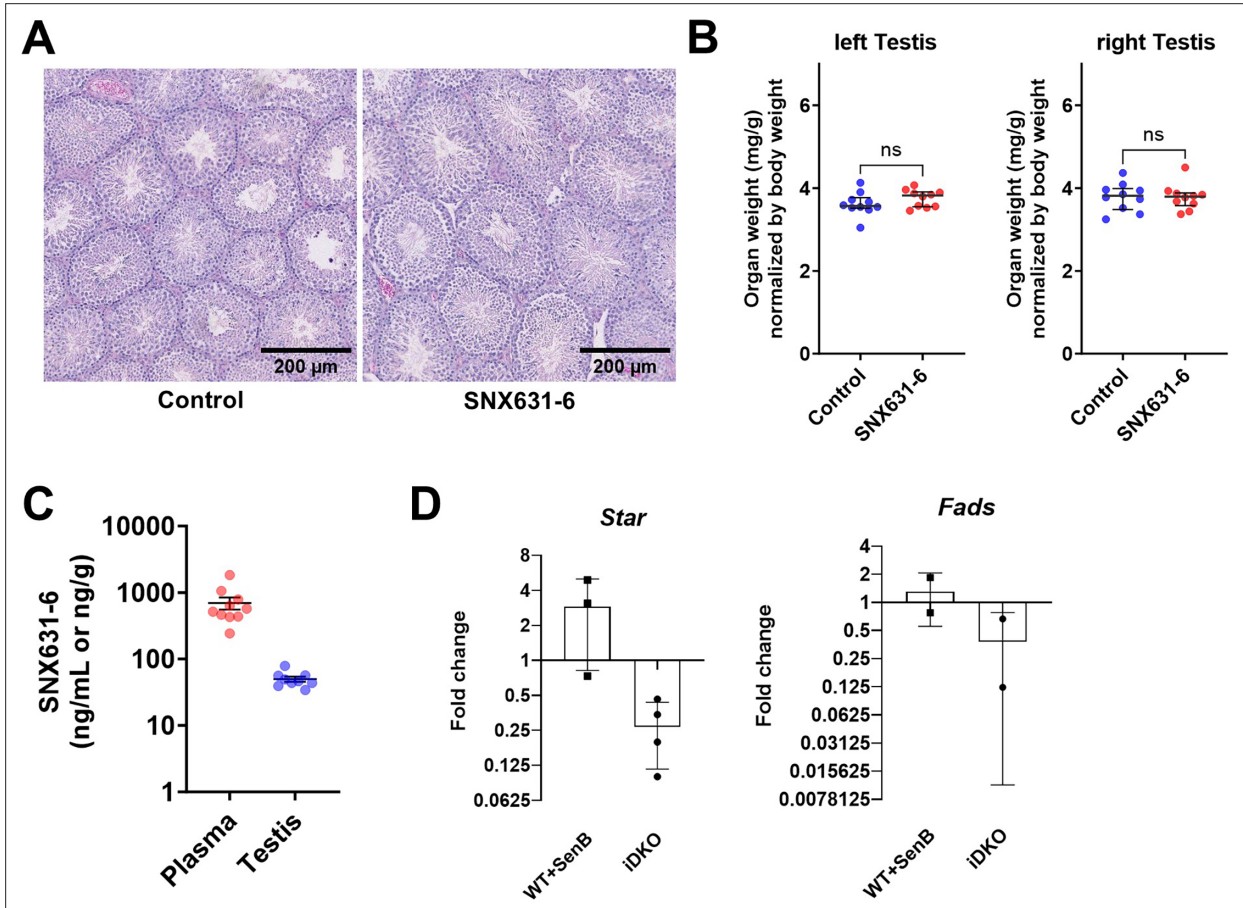

**Figure 8.** The effects of CDK8/19 inhibitor on spermatogenesis in mice. (**A–C**) Male C57BL/6 J mice were treated with SNX631-6 medicated chow (500 ppm, 40–60 mg/kg/d dosage, on average) for 3 wk. (**A**) Representative images of H&E histology analysis of the testicular tissues collected from animals in control or treated groups. (**B**) Organ weights of testes (left and right) at endpoint [unpaired t-test, ns – non significant, n=10]. (**C**) SNX631-6 concentrations in plasma and testicular tissues at endpoint [unpaired t-test, ns – non significant, n=10]. (**D**) qPCR analysis of steroidogenic *Star* and *Fads* genes in ex vivo cultured Leydig cells in response to the CDK8/19 inhibitor Senexin B (1 µM) or hydroxytamoxifen-induced CDK8/19 inducible double knockout (iDKO) [nonparametric t-test, n=3 for *Star* and n=2 for *Fads*].

at a lower level. This corresponds well with the phenotype intensity: CDK19 KO has only decreased fertility, CDK8 iKO are infertile but without evident histological abnormalities, while iDKO exhibit complete spermatogenesis failure. This also corroborates the hypothesis of the CDK8 and CDK19 interchangeability.

Mice with *Med1*<sup>fl/fl</sup>/Vasa-Cre tissue-specific KO had an opposite phenotype to the one described here: prophase I passage was accelerated in these mice and spermatocytes entered zygotene/pachytene stage prematurely (*Huszar et al., 2015*). The difference between these effects is quite interesting in the context of the concept that association with CKM attenuates Mediator-dependent transcription either by sterically preventing TF binding (*Freitas et al., 2022*), or by post-transcriptional regulation of Mediator complex subunits (*Poss et al., 2016*; *Chen et al., 2023*). As far as we know, besides the current work, *Huszar et al., 2015* is the only paper establishing a connection between any components of the Mediator complex and spermatogenesis.

We demonstrated a disruption of steroid biosynthesis in Leydig cells which could play the key role for the observed phenotype. A number of studies (reviewed in *Ilchuk et al., 2023*) affirm a significant role of Mediator complex in lipid metabolism and biosynthesis as one of the main cofactors for the key adipogenic TFs such as C/EBPα, PPARγ, and SREBP.

We detected a decrease in steroidogenic gene expression by scRNAseq (primarily *Cyp17a1*), and then confirmed this finding by WB, immunofluorescence, and direct testosterone measurement. This corresponds well with perturbed Leydig cells morphology (reduced size and secretory vacuoles depletion) and explains the observed atrophy of prostate and epididymis (*Figure 2A*). It is worth mentioning that constitutive *Cyp17a1* knockout in mice leads to female phenotype in XY mice (*Aherrahrou et al., 2020*). Besides *Cyp17a1*, several other steroid metabolism-related genes were downregulated (*Lcn*, *Fads*, *Star*, *Sc5d*, *Sult1e1*). Interestingly, the main feature of *Fads Stoffel et al., 2008* and *Sult1e1* (*Song, 2007*) KOs is male infertility, caused by spermatogenesis arrest.

There is substantial evidence that meiosis in spermatocytes depends on AR signaling (*Wang et al., 2022*). Several papers established a connection between AR signaling, and the Mediator complex (*Russo et al., 2019*). MED1 and MED19 directly interact with the AR and regulate its transcriptional activity (*Jin et al., 2012*; *Weber et al., 2021*). Additional evidence of the role of the Mediator complex and CDK8/19 in the urogenital system comes from studies of their role in prostate cancer, where CDK19 is elevated together with the AR signaling (*Becker et al., 2020*) and where CDK8/19 inhibition and MED12 knockdown partially inhibit the transcriptional AR signaling (*Li et al., 2024*; *Andolfi et al., 2024*).

The iDKO has also dramatically impacted Sertoli cells. The SCARKO mice were engineered two decades ago and show morphological abnormalities (i.e. reduced testes, epididymis, and prostate, meiotic arrest during meiosis I prophase) similar to iDKO (*De Gendt et al., 2004*; *Chang et al., 2004*). The scRNAseq analysis used to elucidate the molecular mechanism of the meiotic arrest revealed a number of similarities with iDKO data for Sertoli cells, especially in terms of pathways (*Cao et al., 2021*). Thus, genes associated with cytoskeleton, cellular membrane, and cell-cell contacts were enriched among the DEGs (*Figure 5B*). Nevertheless, the overlap of genes affected in the SCARKO and iDKO mice was low (*Figure 5—figure supplement 1*). However, an important difference between our model and SCARKO is that *Cdk8* knockout is induced after puberty. Another important consideration is that Sertoli cells are not the only cell type impacting spermatogenesis through AR signaling. AR signaling in peritubular myoid cells is also essential for successful spermatogenesis (*Welsh et al., 2009*). In this regard, probably a more relevant model for comparison with our results is described in *Stanton et al., 2012*, where androgen signaling was suppressed in grown-up rats by low dose testosterone, or its combination with an AR antagonist. In that paper, the rats with the most complete inhibition of AR signaling displayed increased apoptosis in spermatocytes and full blockage of second meiotic division.

Just as in *Stanton et al., 2012*, genes differentially expressed in spermatocytes blocked in prophase I can be divided into the following groups: (1) genes with known roles in meiosis, especially DNA synapse, homologous recombination, and DNA repair, (2) genes associated with cellular stress and apoptosis, and (3) genes with roles in RNA processing and splicing. Changes in the first group can be explained by the compensatory mechanisms and changes in the second group are in concordance with the observed extensive cell death of spermatocytes, whereas changes in the third group remain enigmatic.

Noteworthy, 99.2% of spermatocyte DEGs with |logFc|>0.4 were upregulated genes, which is very similar to the findings of *Pelish et al., 2015* and agrees with the negative regulation of the Mediator complex by CDK8/19 (*Chen et al., 2023*).

Based on these results, we can conclude that iDKO causes perturbance in different testicular cell types: Leydig cells, Sertoli cells, and spermatocytes. We hypothesize that all these changes are caused by disruption of testosterone synthesis in Leydig cells, although, at this point, we cannot definitively prove that the affected genes are regulated by CDK8/19 directly. Data for other cell types are consistent with this hypothesis, however, there is still a possibility that CDK8/19/CCNC have distinct roles in other testicular cell types which are not visible against the background of the testosterone reduction. However, our data are not sufficient to make the conclusion about the role of CDK8/19/CCNC directly in meiosis, besides its hormonal regulation. To answer this question, definitive mouse strains with cell-type-specific iKO in Sertoli cells and meiotic spermatocytes would be needed.

The effects of iDKO on spermatogenesis were maximal at 1–2 mo after tamoxifen administration. At later time points, we detected partial restoration of spermatogenesis, and signs of re-differentiation of the epididymis and activity of secreting cells in the prostate. This restoration did not lead to the production of mature sperm, and the total number of haploid cells increased insignificantly. There are several possible explanations for such an effect. It is unlikely that restoration happened in cells that did not activate Cre-Lox, as the absence of CDK8 has been confirmed by us through PCR (*Figure 1—figure supplement 1B*). Additionally, no restoration of CYP17A1 was found in 5-mo-old animals. There is a possibility that transcription of genes previously regulated by CDK8/19/CCNC was rewired through other transcriptional regulators. An intriguing possibility is activation of a compensatory AR-independent pathway that was recently shown in a zebrafish model (*Zhai et al., 2022*). An argument against such hypothesis would be the absence of such compensation in *Cyp17a1* KO animals or SCARKO mice, but in the former animals the male reproductive system does not develop at all, and in the latter, non-canonical AR signaling through Src kinase may lead to progression to round spermatid stage (*Cooke and Walker, 2021*).

The luteinizing hormone produced in the pituitary gland is a major regulator of steroidogenesis in Leydig cells. It was shown that LH regulates expression level of *Cyp17a1* and several other steroidogenic genes (*Li et al., 2021*; *Ma et al., 2004*). As CDK8 iKO in our model is ubiquitous, infertility could be caused by decrease of LH level due to improper regulation in the brain. However, we measured LH level in the serum and found it unaffected by single or double KOs (*Figure 6D*). That means that CDK8/19/CCNC are required for steroidogenic genes' transcription in Leydig cells, although we cannot definitively prove that CDK8/19 directly regulates the affected genes. And indeed, in our ex vivo experiment with Leydig primary cell culture, even without addition of LH, expression of *Star* and *Fads* was downregulated upon in vitro induction of CDK8 iKO by hydroxytamoxifen on the CDK19 KO background.

Steroidogenesis decline could be related to the altered overall lipid metabolism and precursor deficiency. Several groups reported CDK8/19 role in adipogenesis (*Li et al., 2022*) and lipid biosynthesis (*Tang et al., 2018*). However, contrary to this, we detected high levels of lipids in iDKO Leydig cells (*Figure 6—figure supplement 1*). Similar effect was detected in StAR knockout mice (*Hasegawa et al., 2000*), further confirming that downregulation of steroidogenic gene transcription is the primary cause of steroidogenesis failure in CDK8/19 mice.

The results of the present study reveal for the first time the role of CDK8/19/CCNC in the function of the male reproductive system. The key question regarding the results described here is whether the phenotypic effects are due to the inhibition of CDK8/19 kinase activity or to the loss of CCNC, which is protected by CDK8 and CDK19 from proteasome-mediated degradation (*Chen et al., 2023*), or other kinase-independent functions of CDK8/19. Here, we show that CCNC essentially disappears both from the embryo fibroblasts and from the testes of the iDKO mice. Given the known CDK8/19-independent functions of CCNC (*Ježek et al., 2019*), and the effect of *CcnC* knockout on lipid accumulation (*Song et al., 2022*), the likelihood of CCNC rather than CDK8/19 mediating the observed effects should be investigated. As we have shown, 30 d treatment of 11-wk-old mice with a potent CDK8/19 kinase inhibitor produced no anatomical changes in the testes, and previous reports examining CDK8/19 inhibitors did not produce similar effects. Furthermore, ex vivo treatment of Leydig cells with a CDK8/19 inhibitor did not reproduce the iDKO effect on gene expression, suggesting that the effects of iDKO were kinase-independent and possibly mediated by

CCNC. A definitive answer to the role of CCNC *vs* CDK8/19 kinase activity will require a phenotypic comparison of iDKO mice with mice expressing kinase-inactive mutants of CDK8/19 or inducible CCNC knockout.

## Materials and methods

### Animals and conditional iDKO

*Cdk8*<sup>fl/fl</sup> and ROSA26<sup>CreERT2</sup> (Jax:008463; B6.129-Gt(ROSA)26Sor<sup>tm1(cre/ERT2)Tyj/J</sup>) provenance and genotyping procedures were described previously (*Ilchuk et al., 2022*). *Cdk19*<sup>-/-</sup>, mutant mouse strain C57BL/6N-*Cdk19*<sup>em1(IMPC)J</sup>/Mmucd was obtained from the MMRRC at UC Davis KOMP, RRID:MMRRC_047035-UCD.

Mice were genotyped by real-time PCR using oligonucleotides listed in *Supplementary file 5*. Animals were maintained under controlled room conditions (22–24°C and a 14 hr light: 10 hr dark photoperiod) on standard coniferous (pine) wood shavings bedding with ad libitum access to water and chow approved by the Institutional Animal Care and Use Committee. All studies were conducted in accordance with the principles of biomedical ethics set out in the Helsinki Declaration (1996), approved by the Ethics Committee of the Institute of Gene Biology of the Russian Academy of Sciences (Protocol No. 3 of April 24, 2022) and carried out in accordance with the provisions of Directive 2010/63/EU of the European Parliament and of the Council of the European Union of September 22, 2010, on the protection of animals used for scientific purposes. Tamoxifen treatment was performed as described previously: 8–10 wk-old males were injected with 3 mg of tamoxifen daily for seven consecutive days (*Ilchuk et al., 2022*). Tamoxifen used for KO induction is an estrogen receptor modulator and can by itself affect the mouse organism, especially fertility and hormonal levels (*Willems et al., 2011*). To ensure that the observed effects are caused by the KOs and not by tamoxifen treatment we used tamoxifen-treated controls in every experiment and conducted all experiments but one (*Figure 2C*) at least 1 mo after tamoxifen treatment. Mice were sacrificed by cervical dislocation. To evaluate fertility, males were kept with two CD-1 outbred female mice each. Copulative plugs and newborn pups were monitored daily in the morning.

### Histology

For primary tissue examination, organs were fixed in 10% formaldehyde, paraffin-embedded, sectioned on a microtome and stained with H&E. We used Periodic acid-Shiff (PAS) staining to highlight molecules with a high percentage of carbohydrate content to identify Paneth cells and goblet cells in the ileum.

For immunofluorescence analysis of paraffin sections, the sections were deparaffinized and hydrated using the following steps: 10 min in xylene (twice), 7 min in 100% ethanol (twice), 7 min in 95% ethanol (twice), and 5 min in water at room temperature (three times). Antigen retrieval was performed by autoclaving at 121 °C for 20 min in 10 mM sodium citrate (pH 6), blocked with 5% normal donkey serum in PBS plus Tween 20 (PBST) for 1 hr, then incubated overnight at 4 °C in primary antisera diluted in 1% BSA in PBST. Sections were then washed in PBST and incubated with a 1:200 dilution of secondary antibodies for 1 h (*Supplementary file 5*). Afterwards, the sections were washed in PBST, counterstained for 20 min with 5 ng/mL DAPI (Thermo Fisher Scientific, Waltham, MA), and mounted under coverslips with Prolong Glass (Thermo Fisher).

For immunofluorescent staining, frozen sections of organs were collected in tissue freezing medium (Leica Biosystems, Germany). Sections were cut 3 µm thick in Leica CM1850 cryostat at −20 °C. Slides were incubated for 2 min in 4% PFA and washed with PBS three times for 10 min. Sections were permeabilized in 0.03% Triton-X100 (VWR Life Science, Radnor, PA), 0.1% rabbit serum diluted in PBS at 4°C overnight and washed again with PBS. Sections were stained with primary antibodies (*Supplementary file 5*) at a dilution of 1:100 for an hour, washed in PBS, and incubated with secondary antibodies (1:100). Slides were stained with 5 ng/mL DAPI, mounted with Mowiol 4–88 (Sigma-Aldrich) and examined on Leica STELLARIS confocal microscope (Leica Microsystems, Germany).

For Oil Red O staining, frozen sections (10 µm) were used as described above. Staining was carried out according to the manufacturer's protocol (Sigma, MAK194).

## Flow cytometry

For the isolation of cells from seminiferous tubules we used a modified protocol from *Jeyaraj et al., 2003*. In testicular cells, this method allows one to distinguish between 4 n primary spermatocytes that entered meiosis, 2 n cells (Leidyg, Sertoli, and other somatic cells, spermatogonia stem cells and secondary spermatocytes, that completed first meiotic division) and two 1 n groups: round spermatids and elongated spermatids after DNA compaction, as well as apoptotic (subG1) cells (*Figure 3A*, *left*) (*de Lourdes Pereira et al., 2016*; *Jeyaraj et al., 2003*). The testes of mature mice were placed in phosphate-buffered saline (PBS), the tunica albuginea was opened, and the contents were transferred to a solution of 0.5 µg/mL collagenase IV (PanEco, Russia) in PBS and incubated for 15 min at 32 °C with shaking at 220 rpm. The seminiferous tubules were washed twice with 1 U DNAse (NE Biolabs, Ipswich, MA) in PBS, transferred to 0.01% trypsin (PanEco), and shaken for 15 min at 32 °C, 220 rpm. Trypsin was inactivated with 0.01% bovine serum albumin in PBS; cells were thoroughly resuspended, passed through 70 µm nylon mesh cell strainers (Wuxi NEST Biotechnology, China) and reconstituted in 1 mL PBS. For flow cytometry cells were fixed with 0.75% PFA for 15 min at 37 °C, then washed in PBS. Cells were lysed in the buffer containing 50 µg/mL propidium iodide (PI), 100 µg/mL RNAse A, 0.1% sodium citrate, 0.3% NP-40 (VWR Life Science) for 30 min at 4 °C in the dark followed by flow cytometry analysis on a CytoFlex 26 (Beckman Coulter, Indianapolis, IN) in PE-A and PerCP-A channels. At least 10,000 fluorescents 'events' were collected per sample.

For antibody staining for flow cytometry analysis, cells were fixed with 90% methanol for 30 min and washed three times with PBS. To reduce potential nonspecific antibody staining, Mouse Fc Block (CD16/CD32) (Cell Signaling, #88280) was used as described by the manufacturer. After that, anti-CDK8 (1:100), anti-cyclin C (1:100) and anti-c-Kit (1:20) antibody (*Supplementary file 5*) diluted in 1% BSA in PBS were added. Cells were incubated for 1 hr, washed three times in PBS, incubated with anti-rabbit Alexa-488 conjugated secondary antibody for 1 hr (*Supplementary file 5*), and again washed with PBS three times. For ploidy measurement, cells stained only for CDK8 or CCNC were counterstained with PI as described above (*Figure 2F, G and I, J*, respectively), whereas CDK8 +c Kit or CCNC +c Kit stained cells were counterstained with 5 µg/ml Hoechst 33342 (Invitrogen, H1399) (*Figure 2H and K*) for 30 min at 4 °C in the dark. Flow cytometry measurement using CytoFlex 26. At least 100,000 fluorescent 'events' were collected per each sample.

All the data were analyzed using CytExpert software (Beckman Coulter).

## 10x Chromium library preparation and sequencing

2 R26-Cre-ER^T2 and 2 *Cdk8*^fl/fl*Cdk19*^-/-R26-Cre-ER^T2 tamoxifen-treated male mice at seventh week after activation were sacrificed and tissue cell suspension was prepared as described above. The phenotype was confirmed by cell cycle analysis of a portion of cell suspension used for the 10 X library preparation. Flow cytometry analysis showed a complete absence of haploid cells in one iDKO animal while the other had a subpopulation of round but not elongated spermatids (*Supplementary file 4*).

Single-cell 3' v3 kit and Chromium controller (10 X Genomics) (*Zheng et al., 2017*) were used to generate GEMs for further processing. RT and other stages of single-cell library preparation were performed as per manufacturer instructions, with the exception of AMPure XP Reagent used instead of SPRIselect. These sample libraries were sequenced with Illumina Novaseq6000.

## Single-cell data processing

Raw fastq files generation, alignment and read filtering were done by 10 x Genomics Cell Ranger 6.1.1 with default settings. Read quality was additionally assessed with FastQC. Reads were aligned to the GRCm38 genome, with an average 95% alignment rate. Raw counts, generated by Cell Ranger 6.1.1, were piped to Seurat (*Hao et al., 2021*), where additional filtering was applied. Cells with aberrant UMI vs. genes relations were filtered with pagoda2 genes vs. molecule filter (https://github.com/kharchenkolab/pagoda2; *Barkas and Biederstedt, 2024*) where minimal counts were set as 1000 and minimal genes as 500. Next, cells with high percentages of mitochondrial gene expression per cell were excluded, threshold for filtering set as 25%. After that Scrublet doublet (*Wolock et al., 2019*) score was calculated, cells with scores higher than 0.20 were marked as doublets and excluded. Each sample had more than 5000 high quality cells remaining after filtering. These cells were normalized with the default Seurat method and piped to further filtering and analysis.

To account for ambient gene expression, DecontX (*Yang et al., 2020*) was used, for which cluster-assigned data are required. For stable cluster assignment, a reference dataset was formed of WT samples, which were integrated via Harmony (*Korsunsky et al., 2018*) and then clustered with Seurat. Clustering was checked against the testicular cell type markers (*Hermann et al., 2018*; *Cao et al., 2021*) to ensure selection of true clusters that would reiterate across samples. Cluster of cells belonging to the IGB_2 WT sample only, which did not correspond to any cell type and was missing essential housekeeping genes, was discarded as a cluster of low-quality cells with no biological relevance. This reference dataset was used for automatic cluster mapping with Seurat for all samples, and received labels were used with DecontX to lessen share of ambient gene expression in data.

### Single-cell data analysis

Corrected counts for all cells were piped back to Seurat. Then normalized WT and double knockout samples were integrated with Harmony to account for samples' batch effects. 3480 variable genes were used for PCA and following integration, none of them mitochondrial or ribosomal. Harmony-integrated data was UMAP dimension-reduced and clustered with Seurat. Visualizations were also made with the Seurat package. Cell type was assigned according to their marker genes (*Hermann et al., 2018*; *Cao et al., 2021*). Both cells and microenvironment in the testes were detected in wild-type as well as in KO samples.

Spermatogonia cell cluster was reintegrated and reclustered separately for discovery of spermatogonia subtypes. Integration and clusterization were done as before, with 798 features at the start of the process. Spermatogonia subclusters were assigned types according to literature-curated marker genes (*Hermann et al., 2018*; *Cao et al., 2021*). Differential expression tests (negative binomial Seurat implementation) were conducted for each chosen cell cluster with WT compared against double knockout, batch effect between samples accounted for with the linear model. All mitochondria, ribosomal, and high ambient genes (by DecontX) were excluded from testing. Genes with absolute logFC <0.4 and genes expressed in less than 20% cells in class were also excluded. Additionally, after testing, all genes with no or almost no expression in double knockout, high expressions in WT, and high expression in WT spermatids were excluded as their differential expression might be caused by different ambient composition in WT, heavily influenced by larger spermatid cell proportion. For Sertoli cells, a small subcluster, potentially caused by technical effects, was excluded before testing.

### Assignment of groups and cell count

The clustering analysis was performed according to markers described in previous studies (reviewed in *Suzuki, 2023*). This analysis allowed us to identify nine populations with respective genes: Sertoli cells (*Wt1, Sox9, NR5a1, Clu*), Leydig cells (*Hsd3b1, Cyp17a1*), macrophages (*Cd74, C1qa, Cd45, Ccl5*), T-cells, fibroblasts (*Pdfrgfra, Cd34, Dcn, Gcn*), and the differentiating sperm lineage divided into five clusters: undifferentiated/early spermatogonia (*Crabp1, Tcea3, Kit*), meiotic entry (*Stra8, Rhox13*) leptotene/zygotene (*Sycp3, Dmc1*), two clusters for pachytene (*Piwil1, Id4*) and pachytene/diplotene (*Acr, Pgk2*) (*Supplementary file 4*).

### Hormone measurement

To measure the concentration of hormones in the serum, blood was taken from the left ventricle of the heart, transferred to sterile test tubes, and left at 4 °C overnight. Measurement of luteinizing hormone was performed using ELISA (CEA441Mu, CloudClone, China). The testosterone concentration was measured by the LC-MS method as described previously (*Povaliaeva et al., 2020*). 17α-Hydroxy-progesterone-d8 was used as a standard, proteins were precipitated by adding $ZnSO_4$ and MeOH. The measurement was performed using an on-line extraction method with Agilent Bond Elut C18 cartridges as trap column and Waters Aquity UPLC BEH C18 column, Agilent 1290 Infinity II LC, and AB Sciex Triple Quad 5500 mass-spectrometer.

### Generation of antibodies against CDK19

Coding sequences for aa 377–473 of CDK19 (Q8BWD8) were cloned into pGEX5.1 expression vector. Recombinant GST-tagged CDK19 epitope proteins were purified with Glutathione Sepharose and used for immunization of rabbits to generate target-specific polyclonal antibodies. Animals were immunized every 2 wk with an intradermal injection of a Complete Freund's Adjuvant (Sigma, F5881)

and 500 µg of recombinant CDK19. After 6–8 mo of immunization, blood was collected and antibodies were purified with BrCN-activated Sepharose (Pharmacia).

## Immunoblotting

Protein lysis and western blotting were performed as described in *Ilchuk et al., 2022*. Briefly, cells were lysed in RIPA buffer supplemented with protein inhibitor cocktail (Sigma-Aldrich, St. Louis, MO). Total protein concentration was quantified by the Bradford method. Absorbance at 560 nm was measured with a CLARIOstar Plate Reader (BMG Labtech, Germany). Proteins were separated by SDS-PAGE and transferred onto 0.2 µm nitrocellulose membrane (Bio-Rad, Hercules, CA). After blocking with 5% skimmed milk, membranes were treated with primary antibodies (*Supplementary file 5*) and incubated at 4°C overnight. Membranes were washed with Tris-borate saline with Tween 20 (TBS-T) and incubated for 1 hr at room temperature with secondary antibodies (*Supplementary file 5*). Membranes were visualized with the Clarity Western ECL Substrate (Bio-Rad) using iBright FL1500 Imaging System (Invitrogen, Waltham, MA).

## Leydig cell culture

Seminiferous tubules were isolated as described above for flow cytometry but after collagenase IV (1 µg/mL) digestion, the supernatant was passed through 70 µm nylon mesh cell strainers and centrifuged at 1000 rpm. The cells were resuspended in the medium for Leydig cells (low glucose DMEM (PanEco) with 10% fetal bovine serum (FBS, HyClone, GE Healthcare Life Sciences, Chicago, IL), 1% penicillin/streptomycin (PanEco), 0.3 mg/ml *L*-glutamine (PanEco)) and seeded on Costar 6-well plates (Corning, New York, NY) pre-coated by 0.01% poly-*L*-lysyne (Sigma-Aldrich, P0899), incubated at 37 °C in 5% $CO_2$ for 1 hr and carefully washed with the medium. After 24 hr cells were treated with 0.56% potassium chloride hypotonic solution for 5 min to remove unattached cells. On the seventh day, 1 µM Senexin B (SenB, Senex Biotechnology, Columbia, SC) or 1 µM 4-hydroxytamoxifen (4-OHT, Sigma-Aldrich) was added to the cells. After 7 d of incubation, the cells were lysed with ExtractRNA (Evrogen, Russia).

## qRT PCR

Total RNA from cell suspension was extracted with ExtractRNA (Evrogen) and quantified using Nano-Drop. Equal amounts of RNA were reverse-transcribed to generate cDNA using Superscript II reverse transcription Supermix for qRT-PCR (Invitrogen). qRT-PCR was then performed with SYBR PCR Mix (Evrogen) using the Thermal Cycler system (Bio-Rad). The data thus obtained were analyzed following the comparative (ΔΔCt) method. *Actb* was used as a housekeeping gene. Sequences of primers used are listed in *Supplementary file 5*.

## Statistics

The normality of data was tested using the Shapiro–Wilk test, all datasets met the condition for normality ($p > 0.05$). Two-way analysis of variance (ANOVA) followed by Holm-Sidak's post hoc test for multiple comparisons was used (GraphPad Prism 8; GraphPad Software, San Diego, CA). p-value <0.05 was taken as evidence of statistical significance. 114 mice were used for experiments described here: 34 wild-type and ROSA26[CreERT2] control mice, 16 CDK8 iKO mice, 15 CDK19 KO mice, and 46 iDKO mice. All experiments had at least two biological replicates.

## Acknowledgements

Animal experiments, single-cell RNA sequencing, and histology were funded by Russian Science Foundation project #22-15-00227. Flow cytometry and confocal microscopy experiments were performed on the equipment of the Center for Precision Genome Editing and Genetic Technologies for Biomedicine and funded by the Ministry of Science and Higher Education of the Russian Federation (075-15-2019-1661). We thank Dr. Andrey Yu. Kulibin for providing anti-SYCP3 antibodies and Tamara A Kiriukhina for assistance in isolating the primary MEF culture.

# Additional information

### Competing interests

Mengqian Chen: Director of Research of Senex Biotechnology, Inc. Igor B Roninson: Founder and President of SenexBiotechnology, Inc. The other authors declare that no competing interests exist.

### Funding

| Funder | Grant reference number | Author |
|---|---|---|
| Russian Science Foundation | 22-15-00227 | Alexandra V Bruter<br>Ekaterina Varlamova<br>Nina I Stavskaya<br>Alvina I Khamidullina<br>Alexander A Shtil<br>Victor V Tatarskiy |
| Ministry of Science and Higher Education of the Russian Federation | 075-15-2019-1661 | Alexandra V Bruter<br>Ekaterina Varlamova<br>Anna V Tvorogova<br>Diana S Korshunova<br>Iuliia P Baikova<br>Yulia Y Silaeva<br>Victor V Tatarskiy |

The funders had no role in study design, data collection and interpretation, or the decision to submit the work for publication.

### Author contributions

Alexandra V Bruter, Conceptualization, Supervision, Funding acquisition, Methodology, Writing – original draft, Project administration, Writing – review and editing; Ekaterina A Varlamova, Software, Validation, Visualization, Methodology, Writing – original draft, Writing – review and editing; Nina I Stavskaya, Anna V Tvorogova, Diana S Korshunova, Alvina I Khamidullina, Marina V Utkina, Viktor P Bogdanov, Iuliia P Baikova, Alyona I Nikiforova, Eugene A Albert, Denis O Maksimov, Jing Li, Mengqian Chen, Gary P Schools, Alexey V Feoktistov, Vladislav A Mogila, Methodology; Zoia G Antysheva, Software, Methodology; Vasily N Manskikh, Conceptualization, Methodology; Alexander A Shtil, Igor B Roninson, Funding acquisition, Writing – review and editing; Yulia Y Silaeva, Supervision, Funding acquisition; Victor V Tatarskiy, Conceptualization, Supervision, Funding acquisition, Investigation, Visualization, Methodology, Writing – original draft, Writing – review and editing

### Author ORCIDs

Ekaterina A Varlamova ⓘ https://orcid.org/0000-0002-8451-1216
Mengqian Chen ⓘ https://orcid.org/0000-0003-1706-9509
Igor B Roninson ⓘ https://orcid.org/0000-0002-9211-1327
Vladislav A Mogila ⓘ https://orcid.org/0000-0003-2398-0331
Victor V Tatarskiy ⓘ https://orcid.org/0000-0002-9080-5683

### Ethics

All studies were conducted in accordance with the principles of biomedical ethics set out in the Helsinki Declaration (1996), approved by the Ethics Committee of the Institute of Gene Biology of the Russian Academy of Sciences (Protocol No. 3 of April 24, 2022) and carried out in accordance with the provisions of Directive 2010/63/EU of the European Parliament and of the Council of the European Union of September 22, 2010 on the protection of animals used for scientific purposes.

Reviewer #1 (Public review): https://doi.org/10.7554/eLife.96465.4.sa1
Reviewer #2 (Public review): https://doi.org/10.7554/eLife.96465.4.sa2
Author response https://doi.org/10.7554/eLife.96465.4.sa3

## Additional files

### Supplementary files

Supplementary file 1. Differentially expressed genes (DEGs) associated with Leydig cell lipid metabolism.

Supplementary file 2. GO analysis results for all clusters.

Supplementary file 3. Differentially expressed genes (DEGs) for all cell clusters.

Supplementary file 4. Clusterization markers for main cell types.

Supplementary file 5. List of antibodies and sequences of primers were used.

MDAR checklist

### Data availability

Sequencing data have been deposited in NCBI BioProject under PRJNA1035929 (Run: SRR26671615).

The following dataset was generated:

| Author(s) | Year | Dataset title | Dataset URL | Database and Identifier |
|---|---|---|---|---|
| Institute of Gene Biology, Russian Academy of Sciences | 2023 | Knockout of cyclin dependent kinases 8 and 19 leads to depletion of cyclin C and suppresses spermatogenesis and male fertility in mice | https://www.ncbi.nlm.nih.gov/bioproject/PRJNA1035929 | NCBI BioProject, PRJNA1035929 |

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
