## [Editor Report · eLife Assessment]

This **valuable** study reports the critical role of two cyclin-dependent kinases, CDK8 and CDK19, in spermatogenesis. The data presented are generally supportive of the main conclusion and are considered **solid**. This work may be of interest to reproductive biologists and physicians working on male fertility.

---

## [Referee Report · Reviewer #1 (Public review)]

Summary:

In this paper, Bruter and colleagues report effects of inducible deletion of the genes encoding the two paralogous kinases of the Mediator complex in adult mice. The physiological roles of these two kinases, CDK8 and CDK19, are currently rather poorly understood; although conserved in all eukaryotes, and among the most highly conserved kinases in vertebrates, individual knockouts of genes encoding CDK8 homologues in different species have revealed generally rather mild and specific effects, in contrast to Mediator itself. Here, the authors provide evidence that neither CDK8 nor CDK19 are required for adult homeostasis but they are functionally redundant for maintenance of reproductive tissue morphology and fertility in males.

Strengths:

The morphological data on atrophy of the male reproductive system and arrest of spermatocyte meiosis are solid and are reinforced by single cell transcriptomics data, which is a challenging technique to implement in vivo. The main findings are important and will be of interest to scientists in the fields of transcription and developmental biology.

Weaknesses:

There are several weaknesses.

The first is that data comparing general health of mice with single and double knockouts is not shown, and data on effects in other tissues are sparse and very preliminary. The only strong phenotype of double knockouts that is described is in the male reproductive system. Furthermore, data for the genitourinary system in single knockouts are very sparse; data are described for fertility in figure 1E, ploidy and cell number in figure 3B and C, plasma testosterone and luteinizing hormone levels in figure 6C and 6D and morphology of testis and prostate tissue for single Cdk8 knockout in supplementary figure 1E (although in this case the images do not appear very comparable between control and CDK8 KO), but, for example, there is no analysis of different meiotic stages or of gene expression in single knockouts. Given that the authors have shown that CDK8 and CDK19 expression levels differ widely between different cell types, such an analysis would be interesting. This might have provided insight into the sterility of induced CDK8 knockout.

The second weakness is that the correlation between double knockout and reduced expression of genes involved in steroid hormone biosynthesis is hypothesized to be a causal mechanism for the phenotypes observed. While this is a possibility, there are no experiments performed to provide evidence that this is the case. Furthermore, there is no evidence shown that CDK8 and/or CDK19 are directly responsible for transcription of the genes concerned.

Finally, the authors propose that the phenotypes are independent of the kinase activity of CDK8 or CDK19 because treatment of mice for a month with an inhibitor does not recapitulate the effects of the knockout, and nor does expression of two steroidogenic genes change in cultured Leydig cells upon treatment with an inhibitor. However, there are no controls for effective target inhibition shown.

Comments on revisions:

This manuscript is slightly improved compared to the previous version, though it still does not address the weaknesses that were highlighted in the first version, which largely remain relevant. Please note the typo in the abstract (line 30) and the absence of response to the query of how many crypts and villi were counted in the experiment shown in Suppl Fig 1D.

---

## [Referee Report · Reviewer #2 (Public review)]

Summary:

The authors tried to test the hypothesis that Cdk8 and Cdk19 stabilize the cytoplasmic CcNC protein, the partner protein of Mediator complex including CDK8/19 and Mediator protein via a kinase-independent function by generating induced double knockout of Cdk8/19. However the evidence presented suffer from a lack of focus and rigor and does not support their claims.

Strengths:

This is the first comprehensive report on the effect of a double knockout of CDK8 and CDK19 in mice on male fertility, hormones and single cell testicular cellular expression. The inducible knockout mice led to male sterility with severe spermatogenic defects, and the authors attempted to use this animal model to test the kinase-independent function of CDK8/19, previously reported for human. Single cell RNA-seq of knockout testis presented a high resolution of molecular defects of all the major cell types in the testes of the inducible double knockout mice. The authors also have several interesting findings such as reentry into cell cycles by Sertoli cells, loss of Testosterone in induced DKO that could be investigated further.

Weaknesses:

The claim of reproductive defects in the induced double knockout of CDK8/19 resulted from the loss of CCNC via a kinase-independent mechanism is interesting but was not supported by the data presented. While the construction and analysis of the systemic induced knockout model of Cdk8 in Cdk19KO mice is not trivial, the analysis and data is weakened by systemic effect of Cdk8 loss, making it difficult to separate the systemic effect from the local testis effect.

The analysis of male sterile phenotype is also inadequate with poor image quality, especially testis HE sections. Male reproductive tract picture is also small and difficult to evaluate. The mice crossing scheme is unusual as you have three mice to cross to produce genotypes, while we could understand that it is possible to produce pups of desired genotypes with different mating schemes, such vague crossing scheme is not desirable and of poor genetics practice. Also using TAM treated wild type as control is ok, but a better control will be TAM treated ERT2-cre; CDK8f/f or TAM treated ERT2 Cre CDK19/19 KO, so as to minimize the impact from well-recognized effect of TAM.

While the authors proposed that the inducible loss of CDK8 in the CDK19 knockout background is responsible for spermatogenic defects, it was not clear in which cells CDK8/19 genes are interested and which cell types might have a major role in spermatogenesis. The authors also put forward the evidence that reduction/loss of Testosterone might be the main cause of spermatogenic defects, which is consistent with the expression change in genes involved in steroigenesis pathway in Leydig cells of inducible double knockout. But it is not clear how the loss of Testosterone contributed to the loss of CcnC protein.

The authors should clarify or present the data on where CDK8 and CDK19 as well as CcnC are expressed so as to help the readers to understand which tissues that both CDK might be functioning and cause the loss of CcnC. It should be easier to test the hypothesis of CDK8/19 stabilize CcnC protein using double knock out primary cells, instead of the whole testis.

Since CDK8KO and CDK19KO both have significantly reduced fertility in comparison with wildtype, it might be important to measure the sperm quantity and motility among CDK8 KO, CDK19KO and induced DKO to evaluate spermatogenesis based on their sperm production.

Some data for the inducible knockout efficiency of Cdk8 were presented in Supplemental figure 1, but there is no legend for the supplemental figures, it was not clear which band represented deletion band, which tissues were examined? Tail or testis? It seems that two months after the injection of Tam, all the Cdk8 were completely deleted, indicating extremely efficient deletion of Tam induction by two-month post administration. Were the complete deletion of Cdk8 happening even earlier ? an examination of timepoints of induced loss would be useful and instructional as to when is the best time to examine phenotypes.

The authors found that Sertoli cells re-entered cell cycle in the inducible double knockout but stop short of careful characterization other than increased expression of cell cycle genes.

Overall this work suffered from a lack of focus and rigor in the analysis and lack of sufficient evidence to support their main conclusions.

Comments on revisions:

This reviewer appreciated the authors' effort in improving the quality of this manuscript during their revision. While some concerns remain, the revision is a much improved work and the authors addressed most of my major concerns.

Figure 2E CDK8 and CDK19 immunofluorescent staining images seem to show CDK8 and CDK19 location are completely distinct and in different cells, the authors need to elaborate on this results and discuss what such a distinct location means in line of their double knockout data.

---

## [Author Response]

The following is the authors’ response to the previous reviews.

**Reviewer 1:**
Comments on revisions:This manuscript is in some ways improved - mainly by toning down the conclusions - but a few major weaknesses have not been addressed. I do not agree that it is not justified to perform experiments to investigate the sterility of single CDK8 knockout mice since this could be important and given that the new data show that while there is some overlap in expression of the two prologues, there are also significant differences in the testis. At the least, it would have been interesting and easy to do to show the expression of CDK8 and CDK19 in the single cell transcriptomics, since this might help to identify the different populations.

Certainly, we tried to analyse Cdk8/Cdk19 in single cell transcriptomics. However, we were unable to draw a clear conclusion. Due to a limited sensitivity of single cell sequencing, especially for low abundant transcripts, such as transcription factors (for 10x technology used in our study) (Chuang et al., 2024), it is challenging to establish with certainty CDK8/19 positive and -negative tissues from single cell data because both transcripts are minor. Nevertheless, the majority of cell types showed some expression of CDK8/19, with maximum expression in pachytene/diplotene spermatocytes. We do not include these data to the manuscript particularly as we were successful to assess Cdk8/19 expression patterns using IF approaches.

The only definitive way of concluding a kinase-independent phenotype is to rescue with a kinase dead mutant. While I agree that the inhibitors have been well validated, since they did not have any effects, it is hard to be sure that they actually reached their targets in the tissue concerned. This could have been done by cell thermal shift assay. In the absence of any data on this, the conclusion of a kinase-independent effect is weak.

We totally agree with this point, but it takes several years to produce mice with inducible expression of KD CDK8 mice on the DKO background. These experiments are already underway in our lab, however, their results will be published in our future works.

Figure 2 legend includes (G) between (B) and (C), and appears to, in fact, refer to Fig 1E, for which the legend is missing the description.

Thank you, we corrected this.

Finally, Figure S1C appears wrong. Goblet cells are not in the crypt but on the villi (so the graph axis label is wrong), and there are normally between 5 and 15 per villus, so the DKO figure is normal, but there are a surprisingly high number of goblet cells in the controls. And normally there are 10-15 Paneth cells/crypt, so it looks like these have been underestimated everywhere. I wonder how the counting was done - if it is from images such as those shown here then I am not surprised as the quality is insufficient for quantification. How many crypts and villi were counted? Given the difficulty in counting and the variability per crypt/villus, with quantitative differences like this it is important to do quantifications blind. I personally wouldn't conclude anything from this data and I would recommend to either improve it or not include it. If these data are shown, then data showing efficient double knockout in this tissue should also accompany it, by IF, Western or PCR. Otherwise, given a potentially strong phenotype, repopulation of the intestine by unrecombined crypts might have occurred - this is quite common (see Ganuza et al, EMBO J. 2012).

We added fig. S1C with Western blot showing presence of CDK8 and CCNC in WT intestine and their absence in the DKO intestine. We also corrected that the part of the intestine analyzed was the duodenum, not ileum. We also replaced intestine sections photos with the ones of better quality and higher magnification (200X) and corrected Y axis legend. We apologize for the confusion, and thank the reviewer for careful analysis of our data, which allowed us to make this correction. The numbers of cells were counted on 600x magnification, and the magnification given in the article is for presentation purposes only. Our number of goblet cells was indeed calculated per villus, not crypt, and the resulting number is similar to ones reported in Dannapel et al (Dannappel et al., 2022). As for Paneth cells their numbers correspond to several articles that use the c57bl6 strain (Brischetto et al., 2021; King et al., 2013), as the number of Paneth cells differs between different part of the intestine and different mouse strains (Nakamura et al., 2020).

**Reviewer 2:**
This reviewer appreciated the authors' effort in improving the quality of this manuscript during their revision. While some concerns remain, the revision is a much improved work and the authors addressed most of my major concerns.Figure 2E CDK8 and CDK19 immunofluorescent staining images seem to show CDK8 and CDK19 location are completely distinct and in different cells, the authors need to elaborate on this results and discuss what such a distinct location means in line of their double knockout data.

We thank the reviewer for this suggestion. We had expanded the discussion in the lines 518 and 529 and included a better quality picture of the 200x magnification. Our main line of reasoning is that despite distinct expression in different cell types, high magnification show a certain level of expression of both proteins in most cells, so single knockouts will not demonstrate more than a slight phenotype, while the full knockout will have the full effect. This is especially true if our hypothesis that CCNC stabilization is important here, as both kinases can stabilize the protein.

Minor comments:Supplemental figure 1(C) legend typo : (C) Periodic acid-Schiff stained sections of ilea of tamoxifen treated R26/Cre/ERT2 and DKO mice.

Thank you, we corrected this.

While the effort to identify and generate new antibodies is appreciated, the specificity of the antibodies used should be examined and presented if available.

The specificity of the antibodies for the western blot is confirmed in figure S1F. We added fig. S1G with IF staining of CDK19 KO testes proving our CDK19 antibody specificity.

References:

Brischetto C., Krieger K., Klotz C., et.al. 2021. NF-κB determines Paneth versus goblet cell fate decision in the small intestine. Development 148. doi:10.1242/dev.199683

Chuang H.-C., Li R., Huang H., et.al. 2024. Single-cell sequencing of full-length transcripts and T-cell receptors with automated high-throughput Smart-seq3. BMC Genomics 25:1127. doi:10.1186/s12864-024-11036-0

Dannappel M.V., Zhu D., Sun X., et.al. 2022. CDK8 and CDK19 regulate intestinal differentiation and homeostasis via the chromatin remodeling complex SWI/SNF. J Clin Invest 132. doi:10.1172/JCI158593

King S.L., Mohiuddin J.J., Dekaney C.M.. 2013. Paneth cells expand from newly created and preexisting cells during repair after doxorubicin-induced damage. Am J Physiol Gastrointest Liver Physiol 305:G151–62. doi:10.1152/ajpgi.00441.2012

Nakamura K., Yokoi Y., Fukaya R., et.al. 2020. Expression and localization of Paneth cells and their α-defensins in the small intestine of adult mouse. Front Immunol 11:570296. doi:10.3389/fimmu.2020.570296